# Repurposing the mitotic machinery to drive cellular elongation and chromatin reorganisation in *Plasmodium falciparum* gametocytes

Jiahong Li [1,6], Gerald J. Shami [1,6], Ellie Cho [1,2], Boyin Liu[1], Eric Hanssen [1,3], Matthew W. A. Dixon [4,5] ✉ & Leann Tilley [1] ✉

The sexual stage gametocytes of the malaria parasite, *Plasmodium falciparum*, adopt a falciform (crescent) shape driven by the assembly of a network of microtubules anchored to a cisternal inner membrane complex (IMC). Using 3D electron microscopy, we show that a non-mitotic microtubule organizing center (MTOC), embedded in the parasite's nuclear membrane, orients the endoplasmic reticulum and the nascent IMC and seeds cytoplasmic microtubules. A bundle of microtubules extends into the nuclear lumen, elongating the nuclear envelope and capturing the chromatin. Classical mitotic machinery components, including centriolar plaque proteins, *Pf*centrin-1 and −4, microtubule-associated protein, End-binding protein-1, kinetochore protein, *Pf*NDC80 and centromere-associated protein, *Pf*CENH3, are involved in the nuclear microtubule assembly/disassembly process. Depolymerisation of the microtubules using trifluralin prevents elongation and disrupts the chromatin, centromere and kinetochore organisation. We show that the unusual non-mitotic hemispindle plays a central role in chromatin organisation, IMC positioning and subpellicular microtubule formation in gametocytes.

The most deadly of the human malaria parasites, *Plasmodium falciparum*, is responsible for more than 200 million malaria cases and more than 600 thousand deaths each year[1]. Deaths due to malaria increased by 12% in 2020 as a result of COVID-19-related disruptions to health care, and further increases are feared[1]. These figures exemplify the urgent need to develop strategies that target both the asexual blood stage, responsible for disease symptoms and the sexual blood stage gametocyte, which is responsible for transmission. One potential set of targets are the microtubule networks that play important roles in cell division, motility, invasion and cell shape.

Microtubules are cytoskeletal filaments with a diameter of ~25 nm, composed of alternating rings of α- and β-tubulin dimers. In eukaryotes, microtubules form the scaffolds of the mitotic and meiotic spindles of dividing cells and the axonemes of flagella and cilia[2]. Microtubule-based cytoskeletal networks can provide mechanical support for cells, and they can help position organelles and serve as tracks for intracellular transport of vesicles and mRNA[3]. Microtubules have a slow-growing "minus end", that is often connected into a microtubule-organising centre (MTOC). The growing "plus end" can be dynamic; undergoing growth and retraction[4,5]. Dynamic microtubules

[1]Department of Biochemistry and Pharmacology, Bio21 Molecular Science and Biotechnology Institute, The University of Melbourne, Parkville, VIC 3010, Australia. [2]Biological Optical Microscopy Platform, The University of Melbourne, Parkville, VIC 3010, Australia. [3]Ian Holmes Imaging Center, Bio21 Molecular Science and Biotechnology Institute, The University of Melbourne, Parkville, VIC 3010, Australia. [4]Department of Infectious Diseases, The Peter Doherty Institute, The University of Melbourne, Parkville, VIC 3010, Australia. [5]Walter and Eliza Hall Institute, Parkville, VIC 3010, Australia. [6]These authors contributed equally: Jiahong Li, Gerald J. Shami. ✉e-mail: matthew.dixon@unimelb.edu.au; ltilley@unimelb.edu.au

are capped by β-tubulin and often stabilised by plus end binding proteins[6].

In *P. falciparum*, microtubules play important structural roles in the invasive forms of the parasite and power the motility of flagellated male gametes[7–9]. Microtubules also play critical roles in cell division[10], segregating the daughter cell genomes during nuclear division. The process of mitosis in *P. falciparum* is different from mammalian cells, and more similar to that of other Apicomplexa and fungi. In *Plasmodium*, the nuclear membrane remains intact and the chromosomes are kept in an uncondensed state as nuclear microtubules capture the kinetochores during mitosis[10,11]. An electron-dense acentriolar centrosome (termed the centriolar plaque) is embedded in a pore in the nuclear envelope, and is likely present throughout the cell cycle[10,12–17].

In the early stages of mitosis in asexual blood stage *P. falciparum*, the centriolar plaque duplicates; and spindle microtubules are nucleated from the inner plaque structure, which faces the nuclear lumen. In metaphase, the bipolar spindle microtubules capture the kinetochores, which are bound to the centromeres of the duplicated chromosomes[18]. In the parasite's equivalent of anaphase, the sister kinetochores are separated and moved towards the MTOCs[18]. In the equivalent of telophase, the nuclear envelope divides to form two daughter nuclei without cell division, and the nuclear microtubules remain detectable until the cell undergoes cytokinesis in the final stage of schizogony[13]. As the daughter merozoites develop, an apical prominence is formed close to the centriolar plaque that comprises a tubulin-based apical annulus, called the apical polar ring, and a membrane-bound cisternal structure called the inner membrane complex (IMC) or pellicle[7,13,19]. Two to four microtubules are connected to the apical ring and extend, underneath the IMC, towards the basal end of the cell[13,20].

In the sexual blood stage, intraerythrocytic *P. falciparum* develops through five morphologically distinct forms (stages I – V), over a period of about 12 days to form characteristic crescent-shaped gametocytes[21]. The gametocyte elongates by establishing a network of microtubules underneath a gametocyte IMC, which is related to the structure formed in the merozoite[13,22,23]. The flattened cisternae of the IMC are first evident in stage I gametocytes as a line of connected membrane plates. These gradually expand laterally and lengthways to wrap around the elongating gametocyte[23,24]. The underpinning microtubules are tightly connected to the IMC plates. Unlike merozoites, the gametocytes do not contain an apical polar ring[24]; thus, it remains unclear how the sub-pellicular microtubules are initiated and how the assembly of the IMC is controlled.

In this work, we have used a combination of light and electron microscopy to dissect the origin and organisation of microtubule-based structures in gametocytes. To do so, we generated transfectants expressing fluorescently-tagged components of the microtubule and mitotic machinery. These include Endbinding protein-1 (*Pf*EB1) (a microtubule-binding protein), *Pf*centrin-1 and −4 (centriolar plaque/MTOC markers), *Pf*CENH3 (a centromere marker) and *Pf*NDC80 (a kinetochore marker). Using these tools in combination with Tubulin Tracker, we have identified and characterised an elaborate intranuclear microtubule network in gametocytes that emanates from the nuclear membrane-embedded centriolar plaque. We show that bundles of nuclear microtubules are organised into a hemispindle-like structure that is associated with elongation of the nucleus, reaching a maximum length of about 7 μm in stage II/III of development. In late stage III, this network is disassembled, and in stage IV/V, the nucleus becomes more spherical. Upon depolymerisation of the nuclear microtubules, the sub-pellicular microtubules become polyglutamylated, which may help stabilise and control the microtubule network. We provide evidence that the cytoplasm-facing region of the centriolar plaque serves as a nucleation point for sub-pellicular microtubules. The centriolar plaque also appears to initiate the formation and dictates the positioning of the nascent IMC, via an adjacent

ER extension. In addition to the structural changes, we show that elongation of the nucleus in stage II-III gametocytes is accompanied by a redistribution of the chromatin. Kinetochores capture and reposition the chromatin along the nuclear microtubule bundle before contracting back towards the centriolar plaque in the final stage of development. Finally, we show that treatment with the microtubule depolymerising compound, trifluralin, disrupts both sub-pellicular and nuclear microtubules, which stalls gametocyte elongation, chromatin repositioning and microtubule modification.

## Results

### Gametocytes exhibit sub-pellicular and nuclear microtubules

Tightly synchronised *Plasmodium falciparum* gametocyte preparations were examined by transmission EM (TEM) to determine the origins and organisation of cellular microtubules (Fig. 1a–c, Supplementary Fig. 1). As previously reported[13,22], sub-pellicular microtubules underpin the inner membrane complex (IMC), which in turn underlies the parasite plasma membrane (PPM) and parasitophorous vacuole membrane (PVM) in stage II/III gametocytes (Fig. 1a inset, gold arrowheads). In addition to the subpellicular microtubules, bundles of microtubules are evident in the central region of the nucleus (Fig. 1b, c, Supplementary Fig. 1, purple arrowheads). Close examination of these microtubules shows a densely packed bundle ($17 \pm 2$ (average from $n = 10$ cells)) that arises from an amorphous electron-dense structure associated with the nuclear envelope (Fig. 1c, Supplementary Fig. 1d–g, red arrowheads). Electron dense material containing beads of ~30 nm is observed in some sections, likely representing compacted chromatin material (Fig. 1c, Supplementary Fig. 1a, black arrowheads).

### Nuclear microtubules emanate from a membrane-embedded MTOC

We next employed serial section transmission electron microscopy (ssTEM) to generate 3D ultrastructural maps of stage I to III gametocytes that had been fixed and embedded in epoxy resin. Volume reconstructions were generated, segmented using manual tracing and thresholding methods and rendered automatically, before manual curation of the models (Fig. 1d–f). (See Supplementary Movie 1–3 for translations through the individual sections and rotations of the rendered models). Rendering the parasite surface (green) and nucleus (blue) reveals the remarkable elongation of the nucleus that accompanies the parasite elongation, as previously noted[24,25]. Quantitative analysis of stage I to V gametocytes by Array Tomography (AT) reveals a moderate increase in both surface area and volume of the nucleus during gametocyte maturation. The nucleus reaches a maximum length to width ratio of 4.3:1 at stage III, before contracting back to a more spherical aspect by stage V of development (Supplementary Fig. 2).

The 3D imaging confirms that an amorphous electron-dense MTOC (red) is present at the nuclear envelope (blue) from stage I and that microtubule bundles (yellow) are assembled in stage II/III. The intranuclear microtubule bundles are associated with pointed nuclear protrusions (Fig. 1e, f). Foci of chromatin material (orange) appear to be attached to the microtubule bundle. These 3D EM data can be best appreciated by examining the virtual translations and rotations in Supplementary Movie 1–3.

### Cytoplasmic microtubules initiate at the outer centriolar plaque

To examine the initial stages of formation of both the nuclear and sub-pellicular populations of microtubules at high resolution, we generated 3D volumes by electron tomography. A section from an electron tomogram of a stage II gametocyte (Fig. 1g, h) reveals the MTOC (rendered in red) embedded in the nuclear membrane (rendered in cyan and blue). A translation through the virtual sections is presented in Supplementary Movie 4. Close examination shows that the MTOC

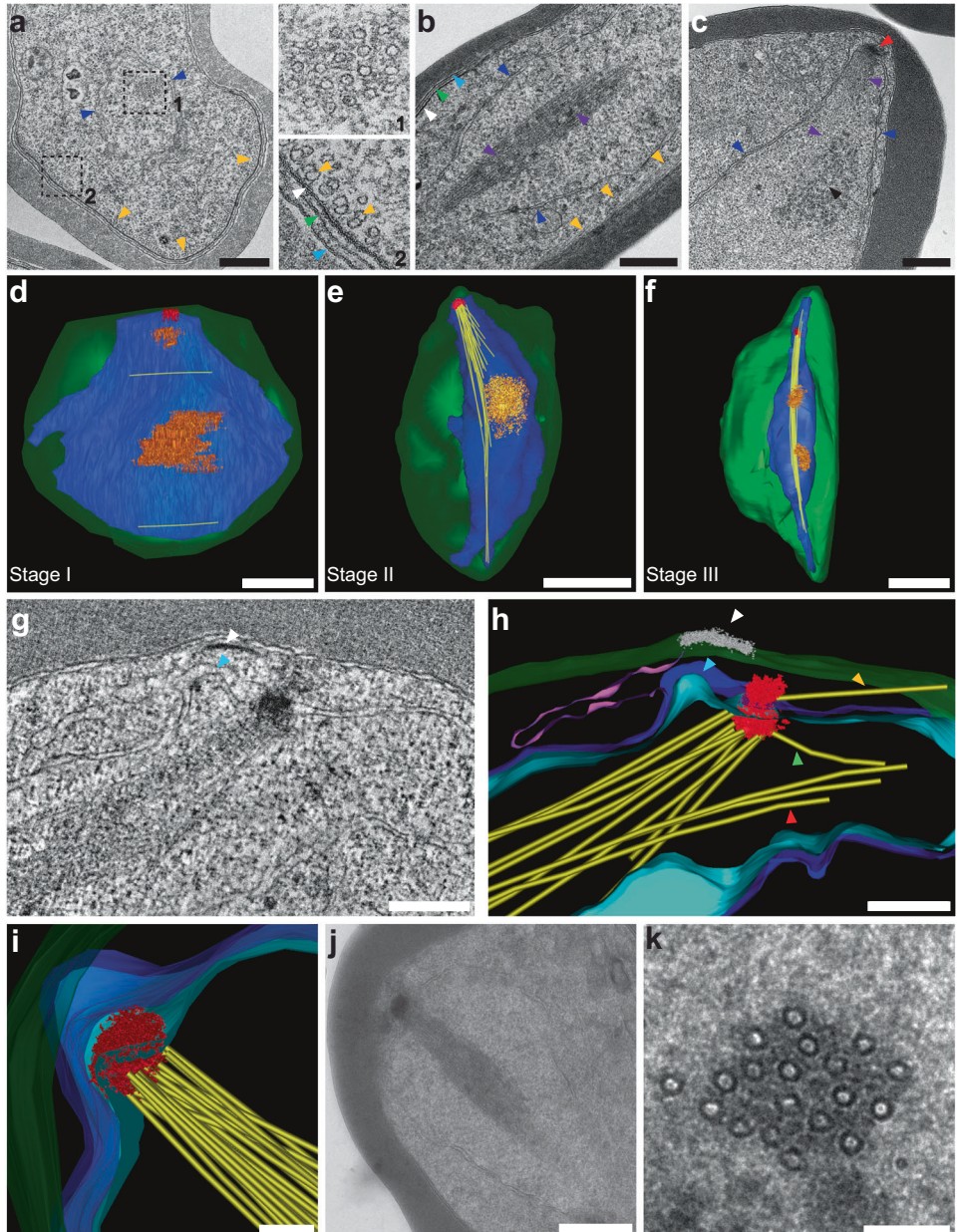

**Fig. 1 | Multimodal electron microscopy reveals intranuclear microtubules in gametocytes. a–c** Thin section transmission electron microscopy (TEM) images of early-stage gametocytes. **a** Stage II gametocyte. Inset 1 shows a transverse section through a bundle of microtubules in the nucleus. Inset 2 shows the sub-pellicular microtubules (gold arrowheads), lying underneath the inner membrane complex (IMC, white arrowhead), parasite plasma membrane (PPM, green arrowhead) and parasitophorous vacuole membrane (PVM, cyan arrowhead). **b** A longitudinal section through a stage III gametocyte illustrating the nuclear microtubule bundle (purple arrowheads) and sub-pellicular microtubules (gold arrowheads). **c** A longitudinal section through a stage III gametocyte illustrating the nuclear microtubule bundle (purple arrowheads) emanating from an amorphous microtubule organizing centre (MTOC, red arrowhead), interacting with chromatin material (black arrowhead). **d–f** Whole cell reconstructions from EM serial sections of stage I-III gametocytes. The plasma membrane (green), nuclear membrane (dark blue), nuclear microtubules (yellow), MTOC (red) and the regions of chromatin (orange) are rendered. Translations through the individual sections and rotations of the models at the three stages are presented in Supplementary Movies 1–3. **g**, **h** Virtual section and rendered model from an electron tomogram, showing the initial stage of the formation of intranuclear microtubules (yellow) at an MTOC (red) in a stage II gametocyte. The MTOC is close to a bulging region of the nuclear membrane (cyan arrowhead) connected to an endoplasmic reticulum extension (magenta) and is located close to the nascent IMC (white arrowheads). **i–k** Rendered model and virtual sections from an electron tomogram of a stage III gametocyte. The MTOC (red) is embedded within the nuclear membranes (dark blue and cyan). Pixel size, x, y, z: **d** 0.86 × 0.86 × 100 nm; **e**, **f** 1.16 × 1.16 × 100 nm. Scale bars: **a–c** = 500 nm; **d–f** = 2 µm; **g** = 300 nm, **h** = 300 nm, **i** = 500 nm, **j** = 200 nm, **k** = 100 nm. Translations through the individual sections and rotations of the models are presented in Supplementary Movies 4 and 5. Figure 1a–c experiments were performed 3 times. Figure 1g–k experiments were performed 3 times (5 tomograms) from 3 different preparations.

has a similar architecture to the centriolar plaque structures that anchor each of the hemispindles in dividing asexual blood stage parasites[10,12,13,15,16]. Centriolar plaque density is observed on either side of the nuclear membranes. From here on, we refer to the material on the nuclear lumen side as the intranuclear plaque and the material on the cytoplasmic side as the outer plaque. In this stage II gametocyte, a loosely packed bundle of nine nuclear microtubules emanates from the intranuclear plaque (Fig. 1h, rendered in yellow). Another microtubule extends from the same intranuclear body plaque but extends at a different angle within the nuclear compartment (Fig. 1h, green

arrowhead). Another bundle of three microtubules appears to have nucleated separately or to have lost contact with the nuclear body (Fig. 1h, red arrowhead).

Further analysis of this tomogram provides insights into the initiation of the sub-pellicular microtubules. A protrusion of the nuclear envelope (Fig. 1g, h, cyan arrowhead) is observed adjacent to the outer centriolar plaque; and a ribbon of endoplasmic reticulum (ER; rendered in magenta) arises from the nuclear envelope. A thickening associated with the closely overlying PPM (Fig. 1g, h, white arrowhead) represents the initial stage of IMC formation. Of particular interest, the electron tomograms reveal the presence of a microtubule in the parasite cytoplasm that emanates from the outer centriolar plaque (Fig. 1h, gold arrowhead). TEM images showing an additional example are presented in Supplementary Fig. 1f, g. This may indicate that the gametocyte subpellicular microtubules are initiated from the cytoplasm-facing region of the centriolar plaque.

A typical example of the centriolar plaque/MTOC and microtubule bundles in a stage III gametocyte is presented as a rendered model in Fig. 1i. Figure 1j, k shows reprojections showing lateral and transverse sections through the centriolar plaque/MTOC and the microtubule bundle. A translation through the virtual sections is presented in Supplementary Movie 5. Segmentation of different features illustrates the position of the centriolar plaque and nuclear microtubule bundle relative to the parasite membranes (PPM/PVM green), the outer nuclear membrane (deep blue) and the inner nuclear membrane (cyan) (Fig. 1i). The electron-dense centriolar plaque (rendered in red) has an average diameter of $216 \pm 26$ nm ($n = 20$) and is embedded in an indentation in the nuclear envelope (Fig. 1i). By this stage, the microtubules (-17) in the bundle exhibit a roughly parallel alignment within the nuclear lumen (Fig. 1i). The nuclear microtubules have the same average diameter as the sub-pellicular microtubules ($26 \pm 3$ nm ($n = 84$) and $27 \pm 4$ nm ($n = 109$), respectively) and are separated by an average distance of $15 \pm 4$ nm ($n = 79$) resulting in an average bundle diameter of $320 \pm 25$ nm ($n = 50$).

### End-binding protein-1 preferentially marks nuclear microtubules

To probe the organisation of nuclear microtubules in live gametocytes, we generated *P. falciparum* transfectants expressing the mCherry fluorescent protein, linked to a nuclear localisation signal (NLS), allowing the nucleus to be visualised[26]. In live stage II and III gametocytes, the NLS-mCherry signal delineates the elongated nuclei, with its pointed extensions that follow the curvature of the parasite (Fig. 2a, b, top panels, yellow arrows). In mature stage IV and V parasites, the nucleus returns to a more spherical profile (Fig. 2a, bottom panels), as observed in the EM imaging.

To visualise the microtubule networks in live gametocytes, we labelled the NLS-mCherry gametocytes with Tubulin Tracker which binds to polymerised tubulin. Tubulin Tracker labels both the nuclear and subpellicular microtubules. The microtubules in the "foot" of the D-shaped stage II/III gametocytes (Fig. 2a, green arrowheads), represent the subpellicular microtubules. A second microtubule population appears to overlap with the NLS-mCherry and Hoechst signals (cyan arrowheads), confirming the presence of the nuclear microtubules in live cells. The microtubule bundle extends in stage III to an average contour length of $7.0 \pm 1.5$ μm (Fig. 2c) before contracting in later stage gametocytes. To our knowledge, antibodies recognising sex-specific markers are not available for early-stage gametocytes; we, therefore, used a large-scale survey of parasites to determine if the nuclear microtubules are present in both male and female gametocytes. We observed nuclear microtubules in 100% of the 249 stage II/III gametocytes examined, indicating that the microtubules are present in both male and female gametocytes.

The plus end-tracking protein, EB1, has been used as a marker of mitotic microtubules in *Toxoplasma gondii*[27]. The N-terminal domain

of the *P. falciparum* homologue of EB1 (PF3D7_0307300) shares 56% similarity with *Tg*EB1, while the rest of the protein, including the putative nuclear localisation signal, is not conserved. Here, we generated transfectants expressing C-terminally tagged *Pf*EB1-GFP expressed from an episomal construct (Supplementary Fig. 3a), in both wild-type parasites and in parasites expressing the NLS-mCherry reporter. Western analysis reveals the expected band with an apparent molecular mass of 84.5 kDa (Supplementary Fig. 3b).

In asexual blood stage parasites, *Pf*EB1-GFP marks punctate structures in the nuclei of dividing schizonts that are closely overlapping with the Tubulin Tracker labelled spindle microtubules; but is not associated with the subpellicular microtubules present in formed merozoites (Supplementary Fig. 3c, Supplementary Fig. 5c, f). In stage II/III gametocytes, *Pf*EB1-GFP is concentrated in the nuclear compartment, as marked by NLS-mCherry (Fig. 2b; Supplementary Fig. 3d); and strongly associated with the population of nuclear microtubules, as marked by Tubulin Tracker (Fig. 2b, cyan arrowheads; Supplementary Fig. 3d, yellow arrowheads). *Pf*EB1-GFP is distributed along the whole length of the nuclear microtubule bundle (Figs. 2b, c, and 3a, Supplementary Fig. 3d), as has been observed previously in *T. gondii*[27]. By stage IV, the sub-pellicular microtubules encase the gametocyte but the population of intranuclear microtubules is no longer evident and the *Pf*EB1-GFP labelling contracts to a single punctum within the nucleus (Supplementary Fig. 3d). In stage V, the sub-pellicular microtubules are also disassembled (Supplementary Fig. 3d).

### Subpellicular microtubules are polyglutamylated

A recent study[16] revealed that the subpellicular microtubules of *P. falciparum* segmented schizonts are polyglutamylated, while the spindle microtubules are not modified[16]. Western analysis using an antibody that recognises polyglutamate (polyE) confirms the signal for polyglutamylated tubulin in late-stage schizonts (Supplementary Fig. 4a), and immunofluorescence microscopy confirmed the differential labelling of the two microtubule populations in schizonts (Supplementary Fig. 4b). Interestingly, neither the nuclear nor the sub-pellicular microtubules are polyglutamylated in stage II gametocytes; however, upon depolymerisation of the nuclear microtubules in stage IV, the sub-pellicular microtubules become polyglutamylated (Fig. 2d, e; Supplementary Fig. 4c, d). Upon disassembly of the microtubule network in stage V, immunofluorescence labelling of anti-β-tubulin shows the presence of remnant monomeric or short chain tubulin oligomers, which remain polyglutamylated (Fig. 2e, Supplementary Fig. 4d bottom panel). Western analysis confirms the polyglutamylation signal in stage IV/V gametocytes (Supplementary Fig. 4c).

### Centriolar plaque initiates nuclear and sub-pellicular microtubules

Centrin-4 has been validated as a marker for the centriolar plaque in *P. berghei*[28] and centrin-1 has been used as a marker in *P. falciparum*[17,29]. We generated transfectants expressing C-terminally tagged *Pf*centrin-4-mCherry or *Pf*centrin-1-mCherry, from episomal constructs, and confirmed protein expression by Western analysis (Supplementary Fig. 5a, b, d, e). In asexual stage parasites, *Pf*centrin-4- and *Pf*centrin-1-mCherry are evident as punctate structures in the nuclei of dividing schizonts that are located at either end of the intranuclear spindles, as indicated by Tubulin Tracker labelling (Supplementary Fig. 5c, f). This confirms that *Pf*centrin-4 and -1 are markers for the centriolar plaque in *P. falciparum*. Interestingly, in daughter merozoites in mature schizonts the *Pf*centrin-1-mCherry signal remains as a concentrated punctum at the end of the subpellicular microtubules (Supplementary Fig. 5c). In contrast, the *Pf*centrin-4-mCherry signal locates adjacent to the chromatin but does not overlap with the Tubulin Tracker signal (Supplementary Fig. 5f).

In developing gametocytes, *Pf*centrin-4-mCherry is visualised as a small concentrated punctum (Fig. 3a), with additional background

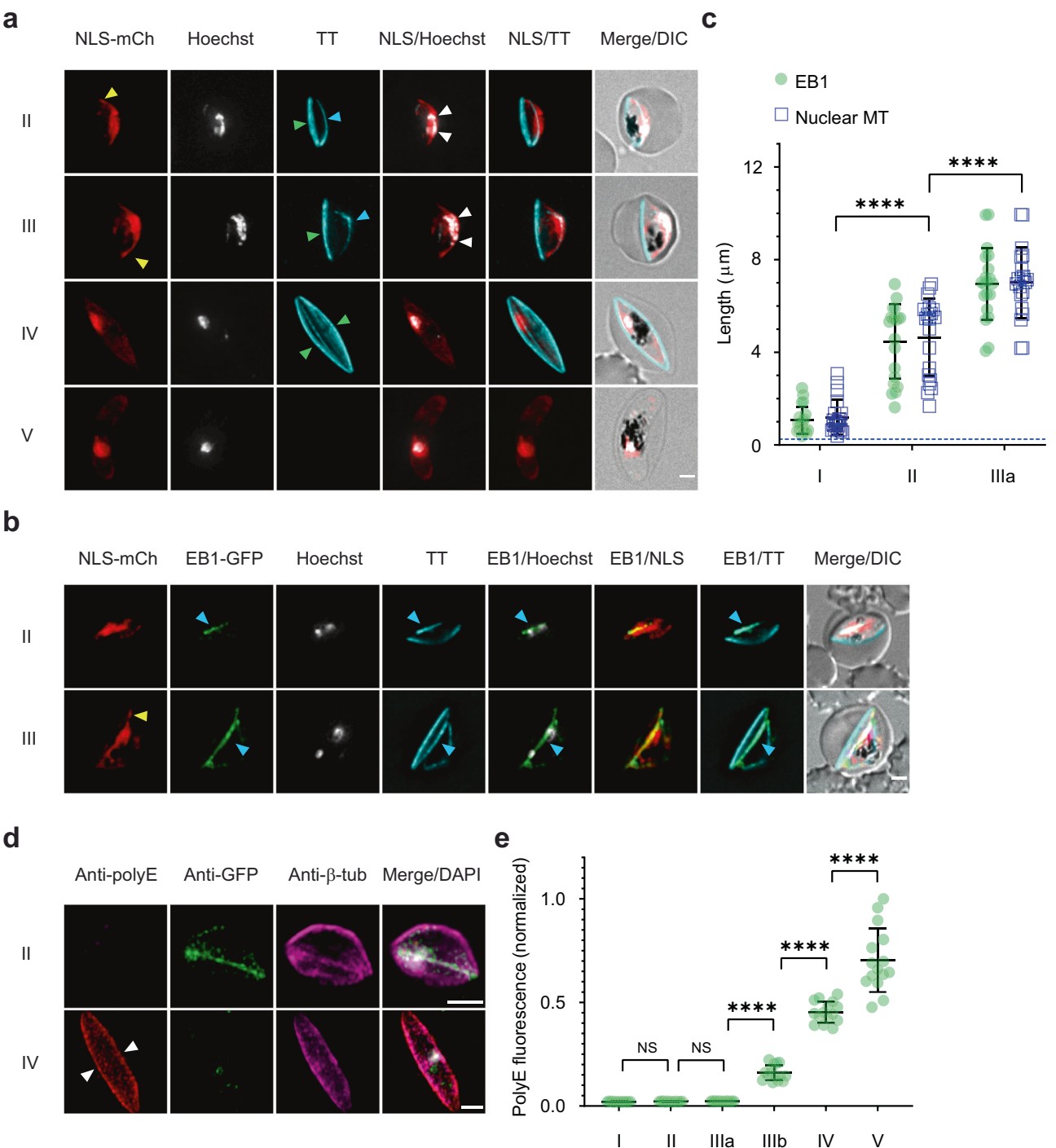

signal throughout the rest of the cell. In the early stages of gametocyte development, when the distortion of the parasite shape is becoming evident (Fig. 3a (stage I-II), magenta arrowheads; Supplementary Fig. 6a, b), the *Pf*centrin-4-mCherry punctum sits at the point where the nuclear and sub-pellicular microtubules are in close apposition. It lies close to the end of the elongating gametocyte, consistent with the location of the centriolar plaque/MTOC observed by EM. Given the spatial arrangement of the centriolar plaque and the nuclear and sub-pellicular microtubules during the early stages of development, we suggest that it may represent a site for seeding both microtubule populations. As the parasite matures, the close apposition of the *Pf*centrin-4-mCherry-labelled structure and the sub-pellicular microtubules is lost (Fig. 3a (stage III), 3b; Supplementary Fig. 6a, b). Interestingly, in a sub-set of the gametocytes (one in six), *Pf*centrin-4-

mCherry labelling is evident as two puncta (Supplementary Fig. 6b, pink arrowheads; Supplementary Fig. 6c). The average distance between the two puncta increases during gametocyte development (Supplementary Fig. 6d).

## Chromatin is captured by microtubule-bound kinetochores

Hoechst 33342 is a cell permeable dye that binds the DNA minor groove in AT-rich regions[30]. It is reported to bind preferentially to condensed regions of heterochromatin; but given the AT-rich nature of *Plasmodium* DNA, it can be considered a general DNA dye in this species. We found that Hoechst labelling is confined to focal regions within the nuclear lumen, as delineated by the NLS-mCherry reporter (Fig. 2a, b, Supplementary Fig. 3d). Of interest, in stage II/III gametocytes, the Hoechst-labelled DNA foci are associated with the nuclear

**Fig. 2 | End-binding protein-1 preferentially associates with the nuclear microtubules, while sub-pellicular microtubules are polyglutamylated. a** Live-cell fluorescence imaging of stage II-V gametocytes transfected with the nuclear localisation signal (NLS)-mCherry reporter. NLS-mCherry (NLS-mCh, red) labels the elongated nucleus, with pointed extensions (yellow arrowheads). Subpellicular (green arrowheads) and nuclear (cyan arrowheads) populations of microtubules are labelled with Tubulin Tracker (TT, cyan). The nucleus contracts to a more spherical morphology and the nuclear and sub-pellicular microtubule populations are disassembled in stage V gametocytes. The chromatin (Hoechst, grayscale) is present as punctate features in the nucleus (white arrowheads). **b** Live-cell fluorescence imaging of stage II/III gametocytes in the *Pf*EB1-GFP/NLS-mCherry co-transfectant parasite line. NLS-mCherry (NLS-mCh, red) delineates the nucleus; and *Pf*EB1-GFP (EB1-GFP, green) is in the same compartment. Chromatin (Hoechst, grayscale) is distributed along the nuclear microtubules. (See Tubulin Tracker (TT, cyan) overlap with *Pf*EB1-GFP, cyan arrows). Additional images are presented in Supplementary Fig. 3d. **c** Quantitative analysis of the contour lengths for the *Pf*EB1-

GFP (EB1) and nuclear microtubules (MT) labelled with Tubulin Tracker (mean ± SD is shown; *n* = 20 cells per stage). Differences in the length of nuclear microtubule features were evaluated by two-way ANOVA Tukey's test, ****$p < 0.0001$. The blue dashed line indicates the limit of resolution (250 nm) of the microscope. **d** Images showing that the nuclear microtubules (as marked by anti-GFP (*Pf*EB1-GFP), green) are depolymerised and the subpellicular microtubules (white arrowheads) are modified with polyglutamate (polyE, red) in stage IV gametocytes. Anti-β-tubulin (anti-β-tub, magenta) labels both nuclear and subpellicular microtubules. The chromatin was labelled with DAPI (grayscale). **e** Quantitative analysis of anti-polyE fluorescence intensity at different stages of development (mean ± SD is shown; *n* = 15 cells). Differences in polyE fluorescence signal were determined using a paired one-way ANOVA Tukey's, I vs. II and II vs. IIIa: NS > 0.9999; IIIa vs. IIIb, IIIb vs. IV and IV vs. V: ****$p < 0.0001$. Differential interference contrast (DIC) images are shown. Scale bars: 2 μm. Additional images are presented in Supplementary Fig. 4d. Source data for Fig. 2c, e is provided in the Source Data file.

microtubules at locations part way along the bundle (see examples in Figs. 2a, b, and 3a; Supplementary Figs. 3d, 6a, b). The capture of compacted packets of chromatin onto the microtubules is consistent with our EM data showing compacted chromatin bound to gametocyte nuclear microtubules at locations away from the centriolar plaque (Fig. 1c–f, Supplementary Fig. 1a). Analysis of the distance between the *Pf*centrin-4-mCherry punctum and the furthermost Hoechst feature (Fig. 3c) reveals that the chromatin shifts away from the centriolar plaque as the nuclear microtubules extend in stage II-III of development, and then is drawn back to the centriolar plaque in stage IV/V when the nuclear microtubules are depolymerised.

The attachment of chromatin to the nuclear microtubule bundle is reminiscent of the process that occurs during mitosis in *Plasmodium*, whereby the kinetochores assemble at specialised chromatin structures on each chromosome, called centromeres, and attach the sister chromatids to microtubule spindles to coordinate their migration into the daughter nuclei. We were interested to determine the locations of the kinetochores and centromeres in these non-mitotic gametocytes. The NDC80 protein forms part of the outer kinetochore complex[31]. We generated transfectants expressing C-terminally tagged *Pf*NDC80-mCherry, from an episomal construct under the control of the native *Pf*NDC80 promoter (Supplementary Fig. 7a). We also generated transfectants expressing the histone H3 variant, *Pf*CENH3 (also known as *Pf*CENP-A; Supplementary Fig. 7a), which is incorporated into the centromeres and has been used previously as a marker for this structure in asexual blood stage *P. falciparum*[32]. Western analysis of these cell lines revealed bands of the expected sizes, confirming the correct expression of the chimeric proteins (Supplementary Fig. 7b).

In schizont stage parasites, *Pf*NDC80-mCherry, GFP-*Pf*CENH3 and the Hoechst-labelled chromatin exhibit profiles that overlap with that of the spindle microtubules (Tubulin Tracker or *Pf*EB1-GFP). In dividing cells, the kinetochore/centromere markers are located between the centriolar plaque puncta, consistent with their roles in attaching chromosomes to the mitotic spindle (Supplementary Fig. 7c–e; Supplementary Fig. 8).

In gametocytes, GFP-*Pf*CENH3 and *Pf*NDC80-mCherry appear to have partially overlapping locations, with an increase in co-occurrence (Mander's coefficient (M)) as the microtubules are disassembled in late stage gametocytes (Fig. 4a; Supplementary Table 1, 2). The Hoechst (chromatin) signal shows substantive co-occurrence with the GFP-*Pf*CENH3 in stage I-III gametocytes; but the overlap decreases as the parasite matures to stage IV/V (Fig. 4a; Supplementary Table 1, 2). Interestingly, there is low co-occurrence of Hoechst with the *Pf*NDC80-mCherry signal in stage I and V; with a moderate increase in stage III (Supplementary Table 1, 2).

As the gametocyte matures from stage I to III, the *Pf*NDC80-mCherry, GFP-*Pf*CENH3 and the DNA signals appear to distribute along the nuclear microtubules (Fig. 4a; Supplementary Fig. 9). In *Pf*centrin-

4-mCherry/GFP-*Pf*CENH3 co-transfectants, the GFP-*Pf*CENH3 is located adjacent to the *Pf*centrin-4-mCherry punctum in stage I gametocytes before moving away during nuclear elongation in stage II/IIIa gametocytes (Fig. 4d, e). Following disassembly of the nuclear microtubules, the GFP-*Pf*CENH3, *Pf*NDC80-mCherry, *Pf*EB1-GFP and *Pf*centrin-4-mCherry punctum again locate adjacent to each other (Fig. 4a–e; Supplementary Fig. 9a, 10). The data suggest that the chromosomes are captured by nuclear microtubule-bound kinetochores and moved away from the centriolar plaque in stage II/III before retracting back in stage IV/V.

## Trifluralin disrupts nuclear morphology and chromatin organisation

Trifluralin is a dinitroaniline herbicide that has been shown to depolymerise *P. falciparum* microtubules[33,34]. Here, we examined the effect of trifluralin treatment (5 μM) on the organisation of microtubule structures. *Pf*centrin-4-mCherry/GFP-*Pf*CENH3 (Fig. 5; Supplementary Fig. 11) and *Pf*NCDC80-mCherry/GFP-*Pf*CENH3 (Supplementary Fig. 12) co-transfectants were committed to gametocytes and treated continuously with trifluralin from Day 0 (gametocyte rings). Live cell microscopy was performed on days 2 to 4 of development. Trifluralin treatment completely ablates the formation of both nuclear and sub-pellicular microtubule bundles, causing loss of the characteristic Tubulin Tracker-labelled bundles, and resulting in a loss of the characteristic elongated shape (Fig. 5a, d; Supplementary Fig. 11a, d, 12a, f, g). On Day 6 of development, parasites were probed with antibodies recognising β-tubulin and polyglutamate. In trifluralin-treated gametocytes, β-tubulin is present but apparently remains unpolymerised and is not glutamylated (Supplementary Figs. 11e, f, 12h). This was confirmed by Western blot analysis of treated Day 7 gametocytes showing an absence of polyglutamylation in trifluralin-treated parasites (Supplementary Figs. 11g, 12i).

In treated gametocytes, the GFP-*Pf*CENH3 (centromere) and Hoechst (chromatin) signals remain closer to the *Pf*centrin-4-mCherry punctum (centriolar plaque) than for control cells, indicating that microtubule bundle formation is required for chromatin reorganisation (Fig. 5a, b; Supplementary Fig. 11a–c). Similarly, both GFP-*Pf*CENH3 and *Pf*NDC80-mCherry fail to redistribute along the nuclear microtubules in treated parasites (Supplementary Fig. 12a–f). These data show that microtubules are needed both for elongation of the gametocyte and for translation of the chromatin away from the MTOC.

## Discussion

Using a series of 3D EM techniques, we have generated a detailed ultrastructural analysis of whole gametocytes at different stages of development. Consistent with previous reports[24,25], we show that elongation of *P. falciparum* gametocytes is accompanied by nuclear elongation and distortion. Here we show that nuclear elongation is

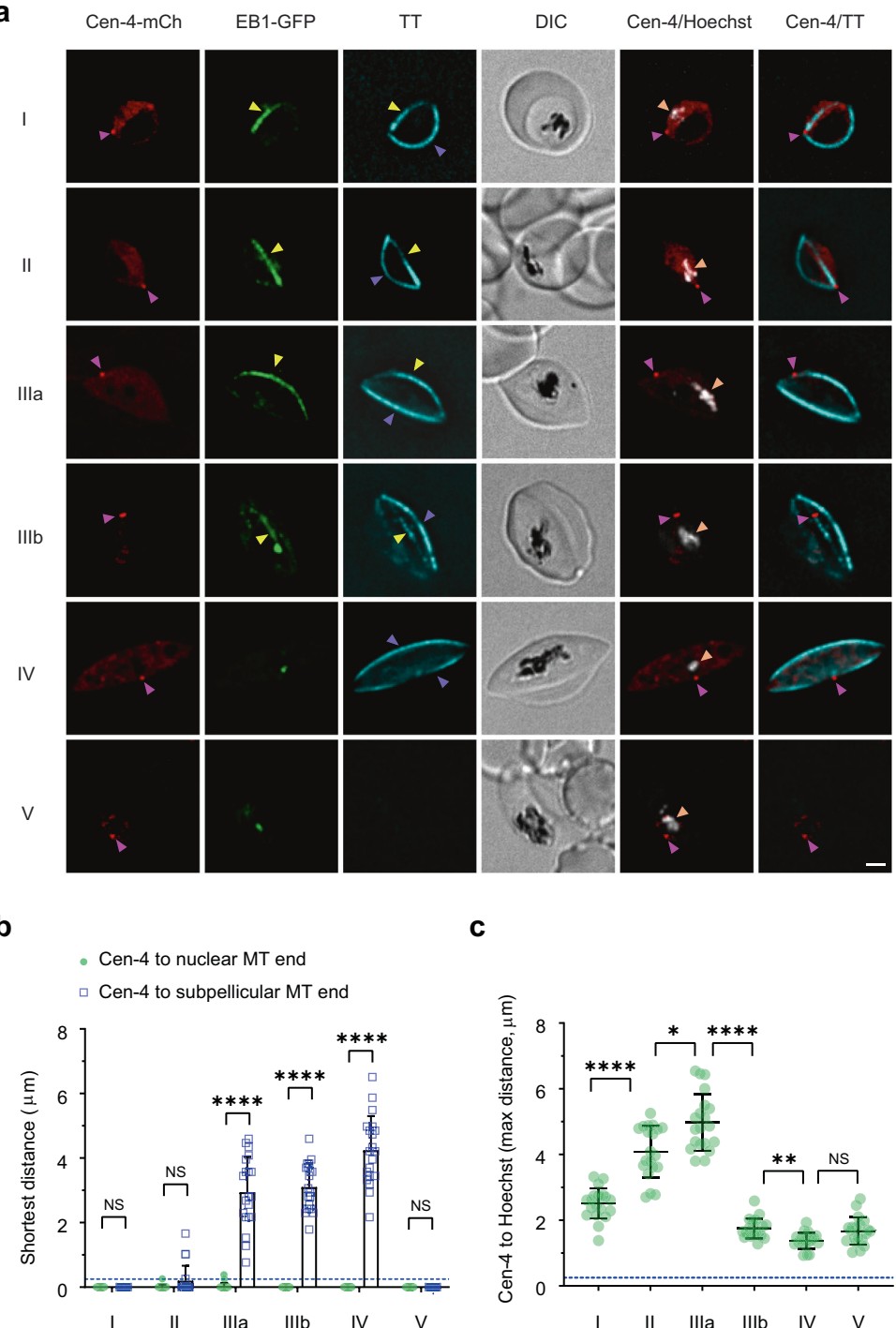

**Fig. 3 | The gametocyte MTOC is a centriolar plaque equivalent. a** Live-cell fluorescence imaging of stage I-V gametocytes in the *Pf*EB1-GFP/*Pf*centrin-4-mCherry co-transfectant parasite line. *Pf*centrin-4-mCherry (Cen-4-mCh, red, magenta arrows) delineates the punctate centriolar plaque and *Pf*EB1-GFP-decorated (EB1-GFP, green)/Tubulin Tracker (TT, cyan)-labelled nuclear microtubules (yellow arrowheads) appear to emanate from this structure in early stage gametocytes (stage I, II). As the gametocyte develops, the sub-pellicular microtubule population (purple arrowheads) appears to move away from the *Pf*centrin-4-mCherry punctum (stage III). In stage IV, the sub-pellicular microtubules form a cage around the gametocyte while the nuclear microtubules collapse. Hoechst (grayscale, orange arrowheads) is used to label chromatin. Differential interference contrast (DIC) images are shown. Scale bars: 2 µm. Additional images are presented in Supplementary Fig. 6a, b. **b, c** Quantitative analysis of the shortest distance between the *Pf*centrin-4-mCherry (Cen-4) puncta and closest ends of the nuclear and subpellicular microtubules (MT) (**b**; mean ± SD is shown, *n* = 20 cells) and maximum distance between the *Pf*centrin-4-mCherry (Cen-4) punctum and the furthermost Hoechst feature (**c**; mean ± SD is shown, *n* = 20 cells). Differences in the shortest distances **b** and maximum distance **c** were determined by two-way and paired one-way ANOVA Tukey's test, respectively. In Fig. 3b, I: NS > 0.9999, II: NS = 0.9918, IIIa, IIIb, IV: ****p < 0.0001, V: NS > 0.9999. In Fig. 3c, I vs. II: ****p < 0.0001, II vs. IIIa: *p = 0.0257, IIIa vs. IIIb: ****p < 0.0001, IIIb vs. IV: **p = 0.0019, IV vs. V: NS = 0.1240. The blue dashed lines indicate the limit of resolution (250 nm) of the microscope. Source data for Fig. 3b, c is provided in the Source Data file.

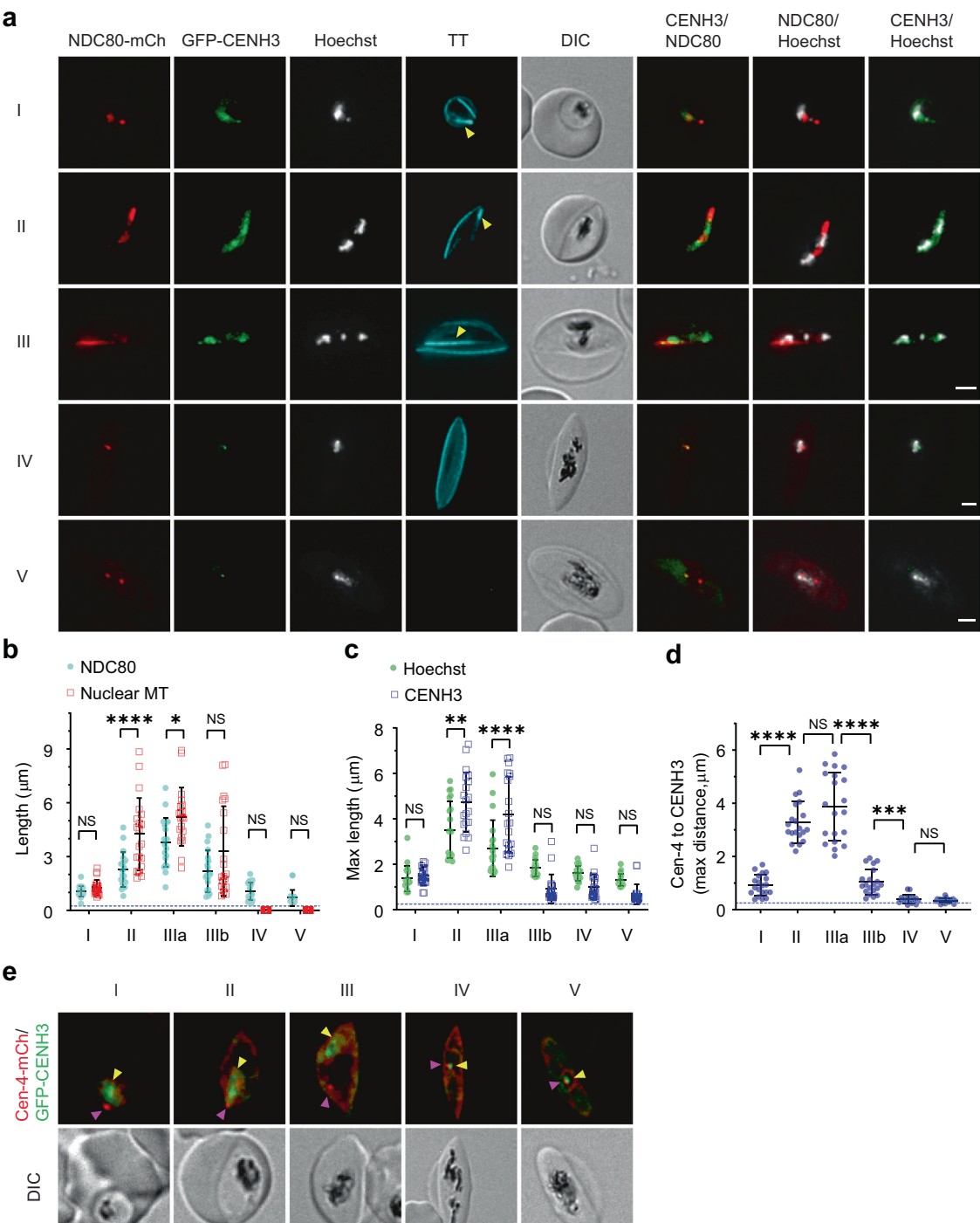

**Fig. 4 | Chromatin is repositioned along the nuclear microtubules in association with the centromeres and kinetochores. a** Live cell imaging of stage I-V gametocyte transfectants expressing *Pf*NDC80-mCherry (NDC80-mCh, kinetochore marker; red), GFP-*Pf*CENH3 (GFP-CENH3; CENH3; centromere marker; green) and counterstained with Hoechst (chromatin; grayscale) and Tubulin Tracker (TT, microtubules; cyan). **b** Contour length of the fluorescence profiles for *Pf*NDC80-mCherry and nuclear microtubules (nuclear MT, marked by Tubulin Tracker with yellow arrows in Fig. 4a) (mean ± SD is shown, *n* = 20 cells). Statistical difference was determined by two-way ANOVA Tukey's test. I: NS > 0.9999; II: ****p < 0.0001; IIIa: *p = 0.0132; IIIb: NS = 0.1481; IV: NS = 0.2092; V: NS = 0.8042. **c** Maximum length for the GFP-*Pf*CENH3 and Hoechst labelling profiles (mean ± SD is shown, *n* = 20 cells). Statistical difference was assessed by two-way ANOVA

Tukey's test. I: NS < 0.9999; II: **p = 0.0011; IIIa: ****p < 0.0001; IIIb: NS = 0.0633; IV: NS = 0.5673; V: NS = 0.5106. **d** Maximum distance between the *Pf*centrin-4-mCherry (Cen-4) and the GFP-*Pf*CENH3 features (mean ± SD is shown, *n* = 20 cells per stage) in the *Pf*centrin-4-mCherry/GFP-*Pf*CENH3 co-transfectants. Statistical difference was analyzed by paired one-way ANOVA Tukey's test. I vs. II: ****p < 0.0001; II vs. IIIa: NS = 0.2030; IIIa vs. IIIb: ****p < 0.0001; IIIb vs. IV: ***p = 0.0004; IV vs. V: NS = 0.6631. **e** Live cell imaging of *Pf*centrin-4-mCherry (Cen-4-mCh, red) and GFP-*Pf*CENH3 (GFP-CENH3, green) transfectants. Differential interference contrast (DIC) images are shown. Scale bars: 2 µm. Additional images and label pairings are presented in Supplementary Fig. 10. Source data for Fig. 4b–d is provided in the Source Data file.

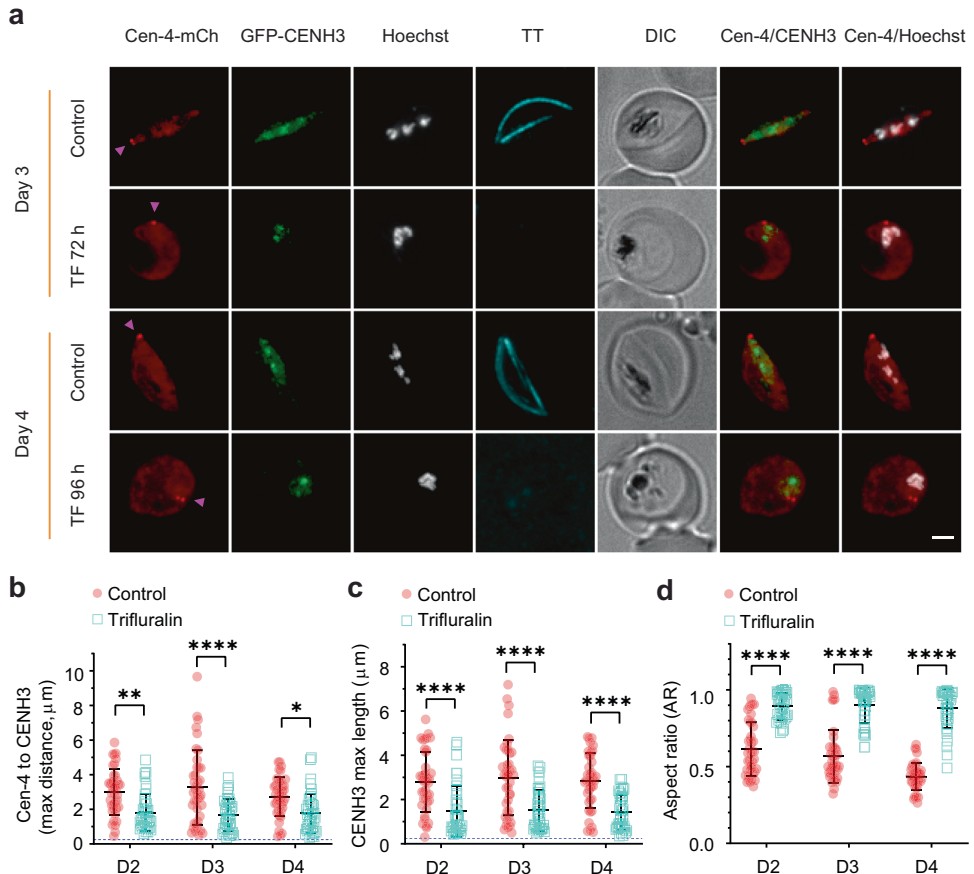

**Fig. 5 | Effects of trifluralin treatment on microtubule organisation and gametocyte morphology. a** Ring stage gametocyte transfectants expressing *Pf*centrin-4-mCherry (Cen-4-mCh, centriolar plaque marker, red, magenta arrowheads) and GFP-*Pf*CENH3 (CENH3, centromere marker, green) were treated with or without 5 µM trifluralin (TF). Images of samples at 72 h (Day 3) and 96 h (Day 4) reveals that both nuclear and subpellicular microtubules are ablated, causing the gametocytes to adopt a round shape. Microtubules are labelled by Tubulin Tracker (TT, cyan) and chromatin with Hoechst (grayscale). Differential interference contrast (DIC) images are shown. Scale bar: 2 µm. A full-time course in presented in Supplementary Fig. 11a. **b** Distance between the *Pf*centrin-4-mCherry (Cen-4)

punctum and the furthermost *Pf*CENH3 feature (mean ± SD is shown, $n = 40$ cells per day). Significance was assessed using a two-way ANOVA Tukey's test, **$p = 0.0015$, ****$p < 0.0001$, *$p = 0.0206$. **c** Max length of the *Pf*CENH3 features (mean ± SD is shown, n = 40 cells per day). Significance was evaluated using a two-way ANOVA Tukey's test, ****$p < 0.0001$. **d** Aspect ratio (width to length) of gametocytes (mean ± SD is shown, $n = 40$ cells per day). Significance was determined by two-way ANOVA Tukey's test, ****$p < 0.0001$. The blue dashed lines indicate the limit of resolution of the microscope. Source data for Fig. 5b–d is provided in the Source Data file.

driven by the formation of bundles of intranuclear microtubules. The microtubule bundles emanate from an electron-dense centriolar plaque/MTOC that spans the outer and inner nuclear membranes. In stage III gametocytes, the nuclear microtubule bundles reach a maximum length of about 7 µm before being disassembled, resulting in contraction of the nucleus back to a roughly spherical shape by stage V of development. Using live cell microscopy of parasites expressing a nuclear targeted NLS-mCherry construct, in conjunction with Tubulin Tracker labelling, we confirmed the presence of the nuclear microtubule bundles.

The fast-growing "plus ends" of microtubule filaments are often stabilised by tip-interacting proteins[6]. EB1 is the only member of the plus end-binding protein family that appears to have a homologue in *P. falciparum*; and *T. gondii* EB1 has been used, previously, as a marker of mitotic microtubules[27]. We generated transfectants expressing *Pf*EB1-GFP and observed association with nuclear, but not sub-pellicular, microtubules. This may be due to the presence of a putative nuclear localisation signal in EB1[27] or may indicate that the nuclear microtubules are more dynamic in nature compared with the highly stabilised sub-pellicular microtubules. *Pf*EB1-GFP binds along the full length of the nuclear microtubule bundles, rather than being restricted to the plus end, consistent with previous data for *Tg*EB1[27].

Our previous work showed that small plates of IMC are formed at the parasite periphery during stage II of development and that these plates act as a scaffold for the assembly of the sub-pellicular microtubule network[24]. In the elongated invasive forms of *P. falciparum*, namely merozoites, ookinetes and sporozoites, the minus ends of the sub-pellicular microtubules are anchored into an apical polar ring[13,35]. Gametocytes do not possess an apical polar ring and the mechanism for initiating and stabilising the minus ends of the sub-pellicular microtubules was not clear. Here we provide evidence that in early gametocytes, both the cytoplasmic and nuclear microtubule populations appear to emerge from the centriolar plaque that traverses the nuclear membrane. In stage IIIb, the nuclear microtubules are disassembled and the centriolar plaque moves away from the sub-pellicular microtubules, which likely are then stabilised by interactions with IMC-associated alveolins[36].

Our EM studies confirmed that the nuclear membrane and centriolar plaque/MTOC come into close contact with the PPM. We observed a region of ER extending from the outer nuclear membrane, adjacent to the nascent gametocyte IMC, which in turn lies just under the PPM. We propose that this represents the first stage of formation of the IMC, which then expands to form the scaffold that underpins the subpellicular network. Super-resolution imaging previously revealed a

close association between the ER and the IMC membranes[23]. The architecture of the nascent IMC in gametocytes is reminiscent of the early stages of merozoite formation in *P. falciparum*[37] and in endomitotic *T. gondii*[38,39], where ER exit sites close to the centriolar plaque structure generate the membrane material for the apical organelles and the IMC. We propose that similar machinery is used to initiate the formation of the gametocyte IMC and to position the subpellicular microtubule network.

In stage IV gametocytes, the nuclear microtubules contract back to the centriolar plaque/MTOC, while the subpellicular microtubules expand to form a cage around the parasite. A recent study[16] revealed that the subpellicular microtubules of *P. falciparum* segmented schizonts are post-translationally modified with the addition of polyglutamates onto the C-terminal glutamate of α- and β-tubulin[40,41]. In contrast, the spindle microtubules are not modified, suggesting that polyglutamylation may be required for subpellicular microtubule stability[16]. Here, we showed that the subpellicular microtubules, but not the nuclear microtubules, are polyglutamylated in stage IV of development. This may facilitate the disassembly of the nuclear microtubules while stabilising the subpellicular microtubule network. In stage V of development, the subpellicular microtubules are dissembled, as judged by the lack of Tubulin Tracker labelling, which binds only to polymerised microtubules. Interestingly, the immunofluorescence signal for both tubulin and polyglutamate remains, perhaps suggesting that the microtubules are reduced to short stabilised stubs. This is consistent with studies in other cell types showing that glutamylation can also be associated with microtubule severing[42].

Trifluralin-treated gametocytes fail to assemble both subpellicular and nuclear microtubules. As a consequence, the gametocyte cell body and nucleus fail to elongate; and the chromatin and kinetochores remain close to the centriolar plaque. Tubulin is present in the treated gametocytes, but appears to remain monomeric or as short stubs and is not modified by polyglutamylation.

Using electron tomography, we found that the centriolar plaque/MTOC structure in gametocytes is similar in appearance to those found in dividing asexual blood stage *Plasmodium* parasites[10]. Interestingly, during male gametogenesis, a bipartite MTOC has been observed that spans the nuclear membrane and connects the centriolar plaque to the axoneme, via proteinaceous material[29,43,44]. This structure is proposed to coordinate mitosis with axoneme formation. Taken with our data, this suggests that Apicomplexan parasites repurpose their centriolar plaque structures at different stages of development to nucleate both nuclear and cytoplasmic microtubules, as required for different cellular events.

Centrins are among the relatively few canonical centrosome components that are conserved in Apicomplexa[28,45]. They are calmodulin-like phosphoproteins that are implicated in centrosome duplication. Transfectants expressing tagged *Pf*centrin-4 exhibit fluorescent foci, from which the nuclear microtubules emanate, again suggesting that the gametocyte MTOC is the equivalent of the centriolar plaque of asexual stage and exflagellating male gametes. In about one in six gametocytes, *Pf*centrin-4 is evident as two puncta. This could potentially represent the bifurcation of the MTOC in male gametocytes ahead of the formation of the basal body[29,43], although further work is needed to test this suggestion.

In canonical mitotic events, one end of the dumbbell-shaped NCD80 complex binds to other centromere-associated kinetochore components, while the other end binds to the plus-end of the microtubules radiating from the spindle. ChiP-seq studies confirmed that *Plasmodium* NCD80 is associated with the chromosome centromeres[46]. We found that *Pf*NDC80 is expressed throughout asexual development, as previously reported for *P. berghei*[31] and *T. gondii*[27]. During gametocyte development, *Pf*NDC80-mCherry is evident as nuclear microtubule-associated foci, close to the Hoechst-labelled chromatin. This is consistent with a recent report

that showed that in male gametogenesis in *P. berghei*, NCD80 exhibits an elongated localisation profile along the microtubules, consistent with kinetochores being distributed along the intranuclear spindle microtubules[31]. Here we show that the centromere-associated modified histone, *Pf*CENH3, is also evident as punctate structures and is associated with the *Pf*NDC80-mCherry-decorated nuclear microtubules. Similarly, Hoechst labelling of live cells and whole cell EM reconstructions revealed localised regions of chromatin, associated with the nuclear microtubules. Taken together, these data are consistent with the capture of the chromosomes onto the nuclear microtubule bundles in gametocytes.

As the gametocytes mature, the *Pf*CENH3 and Hoechst-labelled chromatin structures are moved from a position near the centriolar plaque, to locations along the microtubule bundle in stage II/III, and then back close to the centriolar plaque in stage IV. Thus, the generation of intranuclear microtubules is associated with reorganisation of the captured chromatin. A model illustrating the proposed events is presented in Fig. 6.

The existence of nuclear microtubules and mitotic machinery in a non-replicating parasite stage of the parasite's lifecycle is intriguing and leads us to speculate on their function. It is possible that the formation of the thick bundle of nuclear microtubules contributes to the increased cellular rigidity of the immature (stage II-IV) gametocytes[47,48]. Rigidified gametocytes are mechanically trapped and sequester in low blood flow sites and are not present in the circulation until stage V of development, when the subpellicular microtubule network is disassembled[33,47].

It is interesting to revisit some very early ultrastructural studies of *P. falciparum* gametocytes that provided single section TEM images of intranuclear bodies with a hemispindle appearance, emanating from a region of electron-dense plaque in an enlarged nuclear pore[44,49]. The authors suggested that female gametocytes might undergo a round of endomitosis during development. Molecular characterisation of these structures was not possible at the time and these early studies have been largely forgotten. More recent work[50,51] has shown that gametocytes do not synthesise DNA, and thus remain haploid, until the zygote (the only diploid stage) is formed in the mosquito midgut.

In the absence of DNA replication, it is possible that the capture and repositioning of the chromosomes plays a role in controlling gene transcription. Microtubule-based deformations of the nucleus have been reported to control the gene expression profile in the early stages of hematopoietic differentiation[52]. Chromatin reorganisation may facilitate repression of genes that are required for the early stages of gametocyte development and activation of late stage gametocyte genes. Of interest, short-lived single hemispindles have been described in the asexual blood stage parasites prior to nuclear division[17], suggesting that other stages may also use nuclear microtubules for non-mitotic purposes.

In summary, we have delineated the coordinated assembly and disassembly of a network of sub-pellicular and nuclear microtubules, initiated at an MTOC hub that is embedded in the nuclear membrane of *P. falciparum* gametocytes. The MTOC and associated microtubule network are non-mitotic, instead initiating and co-ordinating IMC formation, driving nuclear and cellular elongation and potentially promoting cellular rigidification. The nuclear microtubules also appear to control the chromatin organisation, which potentially facilitates the regulated expression of proteins at different stages of gametocyte development. An increased understanding of the process of gametocyte biology may point to new tools to arrest the transmission of this deadly parasite.

## Methods

### Parasite culture and production of synchronous gametocytes
Parasite-infected red blood cells (RBCs) were cultured in RPMI-HEPES containing 0.25% AlbuMAX II and 5% human serum[53]. Laboratory strains

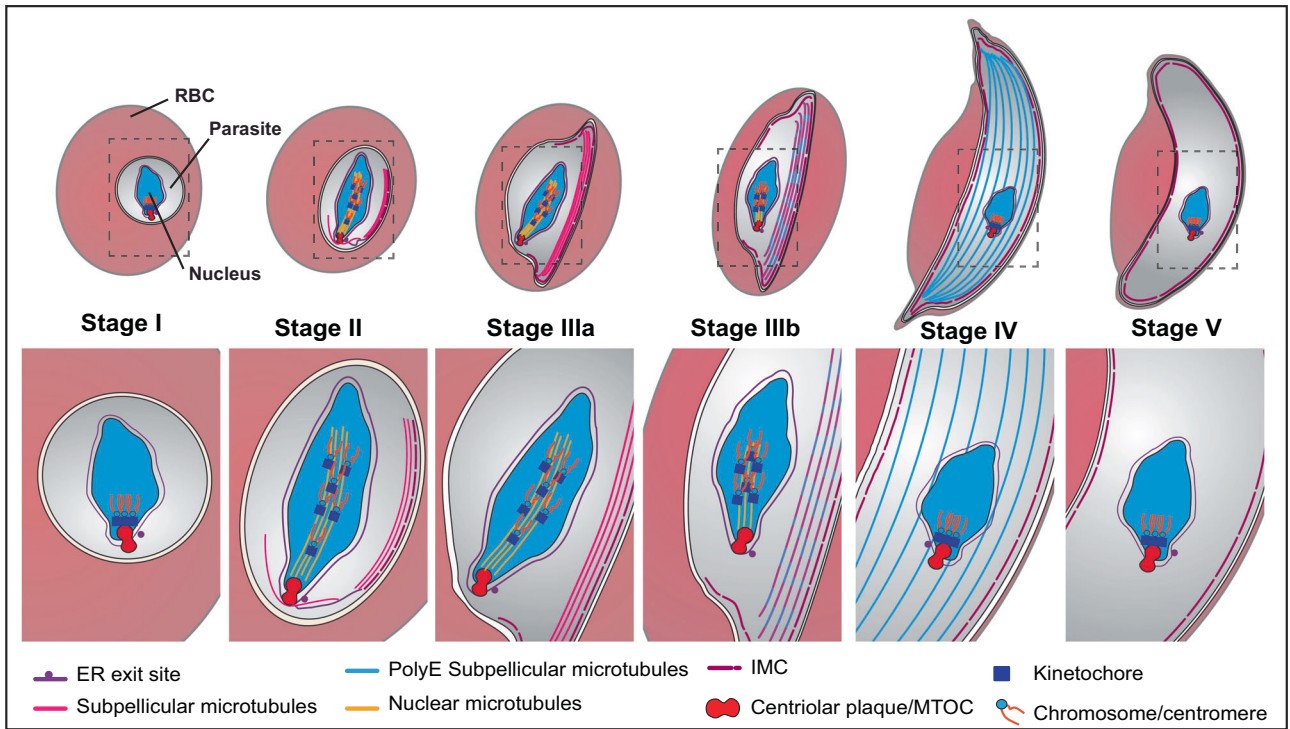

**Fig. 6 | Proposed structural rearrangements of the microtubule networks during gametocyte development.** In stage I, the centriolar plaque, which spans the inner and outer nuclear membranes, initiates the formation of nuclear microtubules. The associated kinetochores capture the centromeres of the chromosomes. In stage II, sub-pellicular microtubules appear to be nucleated from the outer centriolar plaque, and the inner membrane complex (IMC) is initiated from an extension of the nuclear envelope/endoplasmic reticulum (ER) at a site close to the centriolar plaque. As the gametocyte develops, both the intranuclear and sub-pellicular microtubules extend - elongating the nucleus and the whole cell. In stage III, the sub-pellicular microtubules disconnect from the centriolar plaque/ microtubule organizing centre (MTOC) and are stabilised by association with the IMC. In stage III, the kinetochores translate along the nuclear microtubules, thus reorganising the chromatin – an event that may play a role in transcriptional control. In stage IV, the nuclear microtubules are depolymerised to remnant stubs; and the chromatin contracts back towards the centriolar plaque; while remodelling of the subpellicular microtubules is signalled by polyglutamylation (polyE). In stage V gametocytes, the sub-pellicular microtubules are also depolymerised.

of 3D7 kept in culture for long periods form gametocytes with low efficiency. Therefore, we used a 3D7 isolate recovered from a patient and passaged only a limited number of times[54]. To prepare gametocytes[23], a synchronised culture was grown to high parasitemia (~6–8% late asexual stage) and diluted 1:4, retaining the spent culture media. The cultures were maintained for 10–12 days in the presence of 62.5 mM N-acetyl glucosamine to inhibit asexual replication. Gametocyte development was monitored by Giemsa-stained thin blood smears; and gametocytes were harvested at the desired developmental stage.

## Plasmid constructs and transfection

The full-length gene coding sequences for *Pf*EB1, *Pf*centrin-1, *Pf*centrin-4 and *Pf*NDC80 were PCR amplified using the primers in Supplementary Table 3 and cloned into Xho I and Kpn I pre-digested pGLUX or pCLUX plasmids containing either a WR99210-selectable *Hs*DHFR cassette (WR) or a Blasticidin-selectable blasticidin-S-deaminase (BSD) cassette (see Supplementary Table 4). For the *Pf*NDC80 construct, the endogenous promoter was PCR amplified and cloned into the NotI and XhoI sites replacing the CRT promoter sequence. For the *Pf*CENH3 construct, the *Pf*NDC80 promoter was inserted into the Not I and Xho I sites replacing the CRT promoter sequence. The GFP and *Pf*CENH3 sequences were PCR amplified using the primers in Supplementary Table 3, the GFP sequence was digested with Xho I and Kpn I, while *Pf*CENH3 was digested with Kpn I and Pac I. The three sequences were cloned into Not I and Pac I pre-digested pGLUX plasmid containing a WR-selectable marker. Schematics of the constructs are depicted in the relevant supporting figures. The NLS-FRB-mCherry construct was generated previously[26].

One hundred micrograms of plasmid DNA (Supplementary Table 4) were transfected to *P. falciparum* 3D7 ring stage (5% parasitemia) parasites by electroporation[22]. The transfectants were maintained as described above with the addition of a selection drug, namely 5 nM WR, 2.5 μg/mL blasticidin S or 0.9 μM DSM1.

## Trifluralin treatment assays

GFP-*Pf*CENH3/*Pf*centrin4-mCherry or GFP-*Pf*CENH3/*Pf*NDC80-mCherry co-transfectant lines were induced to form gametocytes. The culture medium was supplemented with trifluralin (5 μM) from Day 0 (rings), and the same volume of DMSO was added to the control group. The culture medium was changed daily with the addition of DMSO or trifluralin. Stage distribution analysis was performed on Giemsa-stained slides, counting a minimum of 100 parasitised RBCs. The data presented are the mean percentages of each life cycle stage from 3 separate experiments.

## Western blotting

Parasite-infected RBCs at the desired stage of development were lysed with chilled saponin (0.03% (w/v) in PBS buffer) and incubated on ice for 15 minutes. After centrifugation, the parasite pellets were washed three times with chilled PBS, supplemented with EDTA-free protease inhibitor cocktail (cOmplete). Pellet aliquots (10 μL) were completely resuspended with 25 μL of Bolt LDS sample Buffer (4×) (Invitrogen), 10 μL of Bolt sample reducing agent (10×) (Invitrogen) and 65 μL of chilled PBS buffer. The mixture was boiled at 95 °C for 10 minutes. The samples were loaded into precast Bolt 4 to 12% Bis-Tris protein gels (Invitrogen) and samples were electrophoresed in either MES or MOPS

(Life Technologies) running buffer as per the product instructions. An iBLOT2 system was used to transfer proteins from the gels onto nitrocellulose membranes. The membranes were blocked in 3.5% (w/v) skim milk in PBS containing 0.05% Tween 20 (PBST) for 1 h at room temperature. Primary and secondary antibodies were diluted in the blocking solution. Primary antibodies: rat anti-RFP (red) (ChromoTek; 5f8; 5F8; 1:500), mouse anti-GFP (Roche; 1184460001; clones 7.1 and 13.1; 1:500), rabbit anti-polyglutamate chain (polyE) (AdipoGen Life Sciences; AG-25B-0030-C050; pAb IN105; 1:2,000), mouse anti-β-tubulin (Sigma-Aldrich; T4026; clone TUB 2.1; 1:500), and rabbit anti-*Pf*ERC[55] (1:20,000). Secondary antibodies: goat anti-mouse HRP (Sigma-Aldrich; 619132 (AP181P); 1:10,000), goat anti-rabbit HRP (Sigma-Aldrich; 6KK591 (AP132P); 1:10,000) and goat anti-rat HRP (Invitrogen; A18865; 1:10,000). Blots were incubated with primary antibodies overnight at 4 °C and secondary antibodies for 1 h, at room temperature. The membranes were washed three times with PBST between primary and secondary antibodies and prior to the addition of the clarity-enhanced chemiluminescence reagent (ECL, Bio-Rad) and imaged on a ChemiDoc XRS + gel imaging system (Bio-Rad). Uncropped gels are provided in Supplementary Fig. 13.

## Fluorescence microscopy

For the live cell imaging, Hoechst 33342 (5 μg/mL, Invitrogen) was used to label DNA and Tubulin Tracker Deep Red (1 μM, Invitrogen) marked the microtubules. The cells were incubated at 37 °C for 15 minutes, before mounting onto microscope slides. The images were viewed on a restorative wide field deconvolution microscope (DeltaVision DV Elite, Applied Precision) equipped with a 100× oil immersion objective (numerical aperture (NA) = 1.40). The data were collected and deconvolved using the softWoRx 6.1 software (Applied Precision). The final images were processed to give maximum projections using the FIJI software (version 2.3.0).

For immunofluorescence microscopy, infected gametocytes/ schizonts at the desired stage were aliquoted onto cover slips pre-coated with 0.1 mg/mL erythroagglutinating phytohemagglutinin (PHAE) and incubated for 20 minutes at 37 °C. After washing with PBS, the cells were fixed in 2% formaldehyde in microtubule-stabilizing buffer (MTSB, containing 10 mM 2-(N-morpholino) ethanesulfonic acid (MES), 150 mM NaCl, 5 mM ethylene glycol-bis-(β-amino ethyl ether)-N, N, N′, N′-tetraacetic acid (EGTA), 5 mM glucose, 5 mM MgCl2, pH 8.0) for 30 minutes for gametocytes or 20 minutes for schizonts at 37 °C. The slides were washed three times with PBS and the gametocytes/ schizonts were permeabilised with 0.2% (v/v) Triton-X 100 (in MTSB) for 20 minutes at room temperature, followed by three washes with PBS. The cells were incubated with the primary antibodies for 3 hours at room temperature and washed three times with PBS, followed by another hour with secondary antibodies. The primary antibodies used were: mouse anti-centrin (Sigma-Aldrich; 630249; clone 20H5; 1:200), rabbit anti-*Pf*cenPA (*Pf*CENH3; 1:200)[56], chicken anti-GFP (Abcam; ab13970; 1:200), anti-polyglutamate chain (polyE) (AdipoGen Life Sciences; AG-25B-0030-C050; pAb IN105; 1:200), mouse anti-β-tubulin (Sigma-Aldrich; T4026; clone TUB 2.1; 1:200). The following secondary antibodies were used: goat anti-chicken IgY-Alexa Fluor 488 (Invitrogen; A-11039; 1:500), goat anti-mouse IgG-Alexa Fluor 568 (Invitrogen; A-11004; 1:500), goat anti-rabbit IgG Alexa Fluor 568 (Invitrogen; A-11036; 1:500), goat anti-mouse IgG-Alexa Fluor 647 (Invitrogen; A-21236; 1:500), goat anti-rabbit IgG-Alexa Fluor 647 (Invitrogen; A-21245; 1:500). Parasite chromatin was labelled with DAPI (2 μg/mL in PBS, Thermo Scientific) for 15 minutes at room temperature, followed by washes with PBS buffer. The slides were mounted with the anti-fade solution (90% glycerol/0.2% w/v p-phenylenediamine) before sealing with nail polish. Figure 2d and Supplementary Fig. 4b were acquired using a Zeiss Elyra LSM880 microscope equipped with an Airyscan detector (Carl Zeiss) in "SR" mode and 100× oil immersion objective (numerical aperture = 1.4). The data was processed using Airyscan

processing in the Zen black software (version 2.3) and FIJI software (version 2.3.0) to obtain the maximum projection images. All other images were acquired on the DV Elite microscope as described above. In the case of the images shown in Fig. 4d the native *Pf*EB1-GFP fluorescence was detected and imaged following immunofluorescence labelling and imaging using the DV Elite microscope.

## Data analysis

Fluorescence microscopy images were processed and analysed using FIJI[57]. For anti-polyE intensity measurements, the fluorescence signals of anti-polyE and anti-β tubulin were obtained using identical imaging settings. The multi-plane images were processed using the maximum intensity projection. The mean intensity of anti-polyE fluorescence signals was measured within the cell area, which was identified by the anti-β tubulin channel signal.

Examples of how quantification measurements were made can be found in Supplementary Fig. 14. To quantify the shortest distance from MTOC (*Pf*centrin-4-mCherry) to the nearest end of nuclear micro-tubules (labelled by *Pf*EB1-GFP) or subpellicular microtubules (labelled by Tubulin Tracker), the same threshold and imaging settings were applied in each channel to segment fluorescence signals. The two segmented channels were merged into one panel, and the shortest distance of the combined structure was measured. For analysis of the maximum lengths of Hoechst and GFP-*Pf*CENH3 signals, the same threshold and parameters were set to segment the fluorescence signals in different stage gametocytes, and the Max Feret Diameters (the longest distance between any two points) of the shape were measured via the MorphoLibJ plugins.

To quantify the maximum distance between *Pf*centrin-4-mCherry and Hoechst or GFP-*Pf*CENH3 among different stage cells, the fluorescence signal was segmented using the same threshold and imaging parameters in each channel. Then the two segmented images were merged into the same panel and the Max Feret diameter of the combined structures was measured. For measurements of *Pf*EB1-GFP, *Pf*NDC80-mCherry and nuclear microtubules (labelled by Tubulin Tracker) lengths, identical threshold and imaging parameters were applied to different stages gametocytes. To measure the contour length of each structure, the threshold area of the fluorescence signal was transformed into a thin line (skeletonised) and its Geodesic Diameter was measured using the MorphoLibJ plugin in FIJI.

For the Mander's coefficient (M) analyses, the JACoP plugin was used to manually adjust the threshold to highlight the region of interest. M values were acquired for *Pf*NDC80-mCherry/GFP-*Pf*CENH3, GFP-*Pf*CENH3/*Pf*NDC80-mCherry, Hoechst/*Pf*NDC80-mCherry, *Pf*NDC80-mCherry/Hoechst, Hoechst/GFP-*Pf*CENH3 and GFP-*Pf*CENH3/Hoechst.

## Statistical analysis

All data from experiments are presented as means with error bars defining the standard deviation (SD) as indicated in the figure legends. Fluorescence microscopy-based analyses of gametocyte reporters and Western blots were replicated at least three times. The cell numbers analysed for each experiment are reported in the figure legends. Statistical significance was assessed using one of the following tests: one-way; two-way ANOVA Tukey's test with 95% confidence intervals or *t* test as indicated in the figure legends. Statistical analyses were conducted using GraphPad Prism 8.

## Transmission electron microscopy and electron tomography

Gametocytes at the desired stage were harvested by magnetic separation or Percoll separation. The parasite pellets were resuspended in complete culture medium and incubated at 37 °C for 2 h to allow recovery of microtubules. Infected RBCs were washed in unsupplemented RPMI and fixed in 1 mL of 2% glutaraldehyde (v/v) and 2% formaldehyde (v/v) in 1× PHEM buffer (60 mM PIPES, 25 mM HEPES,

10 mM EGTA, 2 mM MgCl$_2$) for 30 minutes at room temperature (RT). Cells were washed with 1.5× PHEM buffer (3 × 5 minutes) and post-fixed in 1% osmium tetroxide (w/v) in 1× PHEM buffer for 30 minutes in the dark. Cells were washed with 1.5× PHEM buffer (3 × 5 min) and incubated in 1% tannic acid (w/v) in 1× PHEM buffer for 20 minutes. Cells were rinsed in double-distilled water (ddH$_2$O) (3 × 5 min) and incubated in 1% aqueous uranyl acetate (w/v) for 20 minutes in the dark. Cells were subsequently rinsed in ddH$_2$O (3 × 5 min) and dehydrated in an ascending series of ethanol concentrations (30%, 50%, 70%, 90%, 100%, 100%) for 5 minutes each and two changes of absolute acetone for 10 minutes each. Samples were progressively infiltrated with acetone: medium-grade Procure 812 resin (2:1, 1:2) for 2 h, one change of pure resin overnight and a final change for 3 h. Samples were polymerised at 60 °C for 48 h.

Resin blocks were trimmed and sectioned on an ultramicrotome (Leica EM UC7; Leica Microsystems, Wetzlar, Germany), and 70 nm and 250 nm sections were collected on formvar-carbon 100-mesh and single slot copper grids, respectively. Grids were post-stained in 2% aqueous uranyl acetate for 10 minutes, followed by Reynold's lead citrate for 10 minutes. The 250 nm sections were overlaid with 15 nm gold fiducials, which were applied to the backside of grids to facilitate tomogram alignment and reconstruction. Imaging and electron tomography were performed on a FEI Tecnai F30 electron microscope (FEI Company, Hillsboro, OR) at an accelerating voltage of 200 keV. Double-tilt series were acquired for every 2° over a tilt range of ± 60°. Virtual sections were reconstructed from the raw tilt series in Etomo using weighted back-projection algorithms. Image segmentation, visualisation and morphometric analysis were performed using IMOD (version 4.9.9)[58].

### Array tomography
For array tomography (AT)[59], serial sections (120 nm-thick) were generated and collected on a hydrophilised silicon wafer. Sections were post-stained in uranyl acetate and lead citrate as detailed above. Backscattered field emission scanning electron microscopy was conducted using an FEI Teneo operating at an accelerating voltage of 5 kV, beam current of 0.20 nA and working distance of 7 mm. Images were normalised and binned using FIJI[57] (version 2.3.0) and automatically aligned using the StackReg plugin[60]. Segmentation, visualisation and morphometric analysis were performed using IMOD (version 4.9.9) as detailed above.

### Serial section transmission electron microscopy
Serial section transmission electron microscopy (ssTEM) was utilised to generate higher resolution reconstructions of whole cells (stage I-III) relative to array tomography. For each sample, 60–80 serial sections (100 nm thick) were generated and collected onto hydrophilised formvar-carbon copper slot grids. Sections were post-stained in uranyl acetate and lead citrate as detailed above. Sequential images of the same cell of interest were captured using an FEI Tecnai F30 electron microscope at an accelerating voltage of 200 keV. Images were processed and analysed using the same workflow as per array tomography.

### Reporting summary
Further information on research design is available in the Nature Research Reporting Summary linked to this article.

## Data availability
Additional data are available in Supplementary Information. Source data are provided with this paper. The datasets generated and analysed during the current study are available from the corresponding authors on reasonable request. Source data are provided with this paper.

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

## Acknowledgements

We thank Professor Robert Sinden, Imperial College, London, Dr Mike Duffy, University of Melbourne, Dr Jeffrey Dvorin, Harvard Medical School and Dr. Paul McMillan, Peter MacCallum Cancer Centre, for advice. We thank Professor Alan Cowman, Walter and Eliza Hall Institute for providing anti-*Pf*CENH3. We acknowledge the facilities at the Ian Holmes Imaging Centre at Bio21 Institute and the Biological Optical Microscopy Platform (The University of Melbourne). We thank the Australian National Health and Medical Research Council for research support. L.T. was supported by an Australian Research Council Laureate Fellowship.

## Author contributions

J.L., G.J.S., M.W.A.D. and L.T conceived the study. J.L., G.J.S., E.C., B.L. and E.H. performed the experimental work. J.L., G.J.S., E.C., B.L., M.W.A.D. and L.T. processed and analysed the data. L.T. sought funding for the work. J.L., G.J.S., E.C., M.W.A.D. and L.T. contributed to the writing of the manuscript.

## Competing interests

The authors declare no competing interests.
