## [Peer Review File · Nature Communications]

Reviewer comments, first round review

REVIEWER COMMENTS

Reviewer #1 (Remarks to the Author):

Summary:

Understanding the assembly and role of microtubules in the sexual blood-stage parasite of the malaria parasite will potentially lead to a novel strategy to block malaria transmission. In recent years microscopy advances allowed us to discover the conoid in Plasmodium ookinete and the three distinct intranuclear microtubules structures in the replicative blood-stage Plasmodium parasites. The study by Li et al. fills a gap of knowledge by investigating the dynamic of subpellicular and intranuclear microtubules in the sexual blood stage of Plasmodium falciparum parasites. They first employed 3D electron microscopy to characterize the microtubule organizing center (MTOC) and remarkably demonstrated that a single MTOC embedded in the nuclear envelope nucleates both microtubule structures. Next, they generated various transgenic parasites expressing centromeric, centrosomal, or microtubule-associated proteins tagged with fluorescent proteins and monitored their localizations throughout gametogenesis using live fluorescence imaging. The live-cell imaging quality presented in the study is outstanding and allowed the authors to draw robust conclusions about the dynamic of microtubules during the maturation of gametocytes. Although the manuscript could be published in its current state (once major and minor comments are addressed), the study would benefit from reaching higher resolution using ultrastructure expansion.

Major comments:

Introduction:

- Line 51: The authors must add that Tubulin Tracker labeling binds only to polymerized microtubules (information that comes only in the discussion line 389) compared to microtubule visualization using immunostaining with anti-tubulin antibodies.
- Line 61: With the development of advanced microscopy techniques, we have gained tremendous knowledge about the subcellular architecture of the MTOC in the asexual stage of *P. falciparum* recently published in Simon et al. study (Preference number 17 in the current manuscript). However, it remains unknown if the MTOC is "present throughout the cell cycle" and in which form (presence or not of centrin1 protein and DNA-free intranuclear compartment); therefore, I would recommend the authors to soften their language and say it is "likely present...."
- Line 66-67: "In the parasite's equivalent of anaphase, the sister kinetochores are separated and pulled by microtubules towards the MTOCs (Reference: 18). The study cited by the authors confirmed the presence of 14 chromosomes in *P. falciparum* using 3-D reconstruction of EM sections and does not provide any data to test what mechanism drives chromosome separation during Plasmodium blood-stage mitosis. There are several models: (1) "kinetochore microtubules" pull chromosomes, (2) inter-bundle microtubule sliding moves chromosomes, (3) chromosomes slide through lateral interactions with microtubules, or (4) microtubules directly push chromosomes apart. All models have been discussed in a recent publication about mitosis in the *Naegleria* parasite (PMID: 35139359). Because future investigations are needed to address how chromosome segregation occurs during Plasmodium cell division, I would strongly suggest the authors rephrase lines 66-67 to reflect our current knowledge.
- Line 68-70: As the authors were carefully using the accurate term for the MTOC based on previously published data, I would encourage the authors to reword the following sentence by switching the "assembled" for "detectable" so the sentence will read "... and the nuclear microtubules remain detectable until the cell undergoes cytokinesis in the final stage of Schizogony" to not pass on overstatement from previous work published by another research team.
- Line 74: The authors did not reference their statement, which must be updated with the latest literature. In a recent study by Liffner and Absalon (PMID: 34835432), the authors observed subpellicular microtubules with typically 2–4 individual microtubules in each merozoite from

segmented schizonts. More critically to the statement line 74, the authors of this study observed that subpellicular microtubules extend from the apical polar rings, along the length of the merozoite, and end at the basal complex using ultrastructure expansion microscopy combined with NHS ester and tubulin staining.

- Line 82-83: Reference is missing to support the statement that the gametocytes do not contain an apical ring, unlike merozoite.
- Line 93-96: The authors claim that "...nuclear microtubules are organized into a hemispindle-like a structure that drives elongation of the nucleus, reaching a maximum length of about 7 μ m in stage II/III of development," which is not entirely supported by their data. The authors demonstrated in this study that the same MTOC nucleates both nuclear and subpellicular microtubules, and they are both ablated by trifluralin treatment (line 104). Therefore they cannot exclude that a synergistic event could drive the reshaping of the nucleus during gametogenesis due to the extension of nuclear and subpellicular microtubules connected by a common MTOC.

Results:

- Line 148-150: Although I agree with the authors' conclusion and their corresponding references, the authors might want to consider performing an additional experiment by performing ultrastructure expansion microscopy combined with NHS-ester and centrin staining as done in Simon et al. (reference 17) to determine whether the MTOC in asexual and sexual blood-stage parasites harbor an hourglass shaped-structure. This assay will also allow the authors to assess the architecture of the intranuclear compartment of the MTOC (devoid or not of chromatin), to validate the nucleation site of both nuclear and subpellicular microtubules (as the authors beautifully shown by TEM), and to define the organization of the nuclear microtubule (number of branches and length in the hemispindle-like?). Finally, the higher resolution obtained by expansion microscopy will allow redoing their measurement for Cen4- to CENH3 distance if the immunostaining works on expanded parasites.
- Line 296-302: Based on figure 4A, NDC80 and CENH3 signals overlap in stage I and IV and not only in early-stage gametocytes, as mentioned in lines 296-297. It is unclear for what stage of development all Pearson's coefficients were calculated from. For instance, the "r" between NDC80 and Hoechst in stage IV should be close to 1 when it is reported to be 0.251. I recommend the authors show the graph of all Pearson's coefficients for each stage as an additional column on the right of panel A.

Discussion:

- Line 399-400: The authors conclude that the MTOC structure in gametocytes is similar in appearance to those found in dividing asexual/sexual blood-stage parasites. It is true if they had "using electron microscopy."

Simons et al. study, I have nicely demonstrated using NHS-ester on expanded parasites that the asexual MTOC is an hourglass shape composed of two compartments: outer and intranuclear. Moreover, the intranuclear compartment is highly dense in protein devoid of chromatin.

- Line 411-413: The suggestion that the MTOC in male gametocytes bifurcated to form the basal body is exciting but not supported by the data on Pfcenrin-4 immunostaining due to lack of resolution.

Minor comments:

Results:

- Line 240-242: Supplementary Figure 5 C and F, beautiful live-imaging microscopy data demonstrated the localization of EBA1 foci at both ends of different types of intranuclear microtubules (mitotic and extended spindles). To avoid any confusion, I suggest the authors replace the word mitotic with intranuclear "in the nuclei of dividing schizonts located at either end of the mitotic spindles."
- Line 245-247: In the absence of nuclear envelope marker, the authors should say chromatin instead for nucleus in the following sentence: "In contrast, the Pfcenrin-4-mCherry signal locates

adjacent to the nucleus but does not overlap with the TubulinTracker signals (Supp Figure 5F)." Simon et al. (Ref 17) demonstrated that PfCentrin-1 localizes exclusively in the outer nuclear compartment of the MTOC, and additional work is required to address PfCentrin-4's exact localization in the MTOC of asexual blood-stage parasites.

- Line 259-260: The authors' resolution of light fluorescence microscopy is at best 200 nm; therefore, it is technically impossible to be sure that every single punctum of PfCentrin-4-mCherry is single double puncta. "Interestingly, in a sub-set of the gametocytes (one in six), Pfcentrin-4-mCherry labeling is evident as two puncta (Supp Figure 6A, pink arrowheads, B)". I suggest the authors keep the "red" labeled pictures and remove panel B and the text referring to the figure in lines 259-260.
- Line 280: Reference is missing, and it is unclear in which organism the authors are referring to vertebrates, unicellular eukaryotic cells, or parasites?
- Line 334. The reference to figure 5 D is missing in the text

Discussion:

- Line 368: The authors suggest interactions with IMC-associated alveolins stabilize the sub-pellicular microtubules. The alternative hypothesis supported by their data is that the polyglutamylation detected with the anti-polyE could stabilize the microtubules too.
- Line 394: The authors wrote, "Trifluralin-treated gametocytes fail to assemble either sub-pellicular or nuclear microtubules," but they meant "and" (line 222)
- Line 457: References are missing. Simons et al. study demonstrated that hemispindle microtubules do not interact with centromeric proteins using CenH3 as a marker. They also used live-imaging to determine the timing of intranuclear microtubules organization for the first time.
-

General comments for Figures:

Although the authors thoughtfully displayed separate channels in addition to the merged images. I would recommend the authors to rethink the choice of their representative colors for immunostaining and live-imaging regardless of the actual color of the fluorescent protein to make their figures accessible for color-blindness and to increase interpretation for everyone (the cyan color for TT staining made it challenging to interpret merges with green and avoid red, especially in combination with green).

To help the authors see the below-suggested combination of colors:

- Magenta/green/blue
- Magenta/Yellow/Cyan
- Red/Cyan/Yellow

Figure 4

Panel A: Stage V is missing in the panel; could the authors elaborate on why?

Panel B/C/D: The authors might consider drawing a horizontal line around 0.250 μm , marking the resolution indicating that measurements below the line two foci distant of 250 nm or lower will be seen as one focus.

Figure 5:

For Panel B/C/D: Same comments as for Figure 4

In addition, the authors might consider adding a schematic on the top of each graph to explain what is "contour length" and "Aspect ratio"? It will increase the likelihood for another lab to reproduce the study's data if there are no ambiguities on how the cellular measurements were done.

Figure 6:

I compliment the authors for creating a data/model figure. I have a few suggestions. To avoid the combination of Red/green, the parasite could be gray. The column with the RBC present could be smaller to gain space for an additional zoom on the parasite drawing to see better the details in the drawing.

The caption could take the entire width of the figure and place it in the bottom to free space for zoom on the parasite like done for other stages.

Supp Figure 03:

Panel B: The loading control for the 3D7 sexual stage is pretty low, and therefore it is difficult to conclude anything regarding EB1 expression level.

Supp Figure 04:

Panel D: as for panel B, it should read anti-GFP and not EB1-GFP

In my understanding, all figures from panels B and D are from immunostaining, therefore " or endogenous PfEB1-GFP labeling" in lines 68-69 should be removed

Supp Figure 05:

Panel B: Could the authors flip the blot picture vertically, starting with 3D7 samples first as for all other blots in other figures and Panel E?

Supp Figure 6:

A title reflecting the data rather than the experimental design would be best.

Supp Figure 11:

Panel G: The loading control is missing for the anti-PolyE blot.

Supp Figure 12:

The title reads, "Trifluralin treatment..... disrupts cellular remodeling, as...." However, it is vague and suggests the author write something like "chromatin spatial organization" instead.

Reviewer #2 (Remarks to the Author):

The manuscript present by Li et al from Leann Tiley's group is a very illuminating and interesting study illustrating how some of the components of mitotic machinery are used by the human malaria parasite *P. falciparum* in the cellular shape and chromatin reorganisation. *P. falciparum* undergoes various cellular organisation changes during its gametocyte development. The authors show how the cytoplasmic and nuclear microtubule orient in these different stages and how the microtubule organising centre is organised during these different stages even if they are not undergoing mitosis. This is the originality of this work. The authors have used some cutting edge cell biology techniques like 3D electron microscopy and real time live cell imaging which are quite challenging to do in the human parasites. The different markers used in the study are centrosomal marker like centrin 4 and 1, kinetochore marker Ndc80 and microtubule end binding protein EB1. Some of these markers have been studied during mitotic stages by our group and by others (Roques et al 2018; Zeeshan et al 2020; Simon et al 2021, Raspa and Brochet 2022) and hence that is not surprising what they observe but since the study is focussed on the non-mitotic cell and hence it is original in that aspect. The study is bit diluted by the recent study by Raspa and Brochet, *PLoS Pathogens* 2022, where they have shown that microtubule localisation at the sub-pellicle region of the human parasite using expansion microscopy. However saying that, the live cell imaging of the Ndc80 and EB1 is exemplary in this manuscript and gives a better picture of the mitotic machinery in this study. This will be very useful for the community interested in the mitosis and sexual development both in human parasite but also to scientist working on the other pathogens belonging to Apicomplexa and other showing atypical microtubule dynamics.

The authors also substantiate their study by using trifluralin by depolymerising the microtubules and demonstrate that by this treatment blocks the chromatin and kinetochore organisation and thereby IMC positioning in gametocytes.

The study is solid though some of these molecules have been studied in depth before as mitotic markers. It would have been interesting if some of functional data on some these molecules were attempted to show if they affect the gametocyte organisation and at what stages do they affect it. As it has been shown that centrin 4 is not required for asexual stage multiplication and hence it will interesting to know if the centrin 4 affect the non-mitotic assembly during gametocyte. The same will be informative if it was demonstrated what happens with EB1 as in *Toxo* it is not required for asexual multiplication.

I really enjoyed reviewing the manuscript and it is an excellent study and some outstanding

multimodal electron microscopy and real time live cell imaging is present for gametocyte. There is lot of data in the supplementary about the asexual stages which is very informative for the community. Overall this study will be of great interest to the scientist working on MTOC and mitosis in Plasmodium and other non-model eukaryotes.

Rita Tewari

Reviewer #3 (Remarks to the Author):

In this manuscript Li and colleagues describe an enigmatic structure resembling a mitotic spindle during Plasmodium micro- and macro-gametocytogenesis, two developmental stages that are not known to undergo mitotic division. However, three rounds of endomitosis quickly happen following microgametocyte activation by a decrease of temperature of at least 5C. In the absence of genome replication and mitosis, the authors propose that this process is used by gametocytes to drive their characteristic cellular elongation and to reorganise chromatin. To describe this structure, the authors combine various imaging approaches including 3D electron microscopy and fluorescence microscopy with known markers of the mitotic spindle, the kinetochore or the centromere.

The idea that Plasmodium falciparum gametocytes use the mitotic machinery for cell elongation and chromatin organisation is tantalizing and would represent an important discovery. Gametocytogenesis attracts a growing interest but has also been well scrutinised by electron microscopy and such structure has never been observed nor described, which is relatively surprising. Therefore, I think that the authors should make 100% sure they are not observing an artefact of premature activation of microgametocytes due to unwanted temperature drops, which are in our hands frequently observed particularly at pH >7.2. In the comments below we note some observations that raise some concerns and propose experiments to address them.

Major comments

In figure 1 it is not clear how many cells were analysed. Could the authors use known markers of male and female gametocytes to ascertain that their observations hold true for both sexes?

In figure 2, could the authors confirm with an ER marker such BiP the proposed correlation between the endoplasmic reticulum, the nascent IMC and the microtubule organisation? It would be important to quantify particularly if the number of cells analysed in figure 1 is limited.

It would be essential to confirm that the observations made in figure 2 hold true for both male and female gametocytes using specific markers. The authors show parasites expressing NLS-mcherry. However, it seems that that expression of this reporter protein is under the control of the hsp86 promoter. According to transcriptomics studies available on PlasmoDB, hsp86 promoter is up to 10 times more active in microgametocytes. It is thus possible that the authors have been focusing their analysis on microgametocytes. Similarly, in figure 3 and 4 it would be essential to ensure that the measurements hold true for both macro- and microgametocytes.

Finally, if only microgametocytes show this structure, the authors should ensure that this observation is not linked to unwanted premature and suboptimal activation of microgametocytes due to experimentally inherent fluctuations in temperature. It was previously shown that in Plasmodium falciparum, CDPK4 is essential for the formation of the mitotic spindle during microgametogenesis but that mutant gametocytes lacking *cdpk4* are nevertheless elongated (Kumar et al, 2021, mbio). I would strongly recommend to the authors to document the formation of these structures resembling a mitotic spindle during gametocytogenesis of mutants lacking *cdpk4* or in presence of an inhibitor of CDPK4 such as compound BKI1294. This should rule out that the authors are describing an experimental artefact.

Minor comments

In figure 4, it is not clear how the authors define the relative movement of Ndc80, CenH3 to the centriolar plaque(s) in the absence of a centrin staining and with a bimodal distribution of CenH3 and Ndc80.

The authors propose that this microtubule spindle is important for the elongation of the cells. The images shown here suggest that the orientation of the spindle is indeed sagittal to the longest length of gametocytes but this is not quantified. Could it also be a consequence of elongation rather than a cause?

A link between mitotic MTOC and cell polarity has already been described in *Toxoplasma* (see for example the review from Harding and Frischknecht (Trends in parasitology, 2020) and would be worth mentioning.

Response to reviewers

Reviewer #1

Summary:

Understanding the assembly and role of microtubules in the sexual blood-stage parasite of the malaria parasite will potentially lead to a novel strategy to block malaria transmission. In recent years microscopy advances allowed us to discover the conoid in Plasmodium ookinete and the three distinct intranuclear microtubules structures in the replicative blood-stage Plasmodium parasites. The study by Li et al. fills a gap of knowledge by investigating the dynamic of subpellicular and intranuclear microtubules in the sexual blood stage of Plasmodium falciparum parasites. They first employed 3D electron microscopy to characterize the microtubule organizing center (MTOC) and remarkably demonstrated that a single MTOC embedded in the nuclear envelope nucleates both microtubule structures. Next, they generated various transgenic parasites expressing centromeric, centrosomal, or microtubule-associated proteins tagged with fluorescent proteins and monitored their localizations throughout gametogenesis using live fluorescence imaging. The live-cell imaging quality presented in the study is outstanding and allowed the authors to draw robust conclusions about the dynamic of microtubules during the maturation of gametocytes. Although the manuscript could be published in its current state (once major and minor comments are addressed), the study would benefit from reaching higher resolution using ultrastructure expansion.

We appreciate the encouraging comments from the reviewer; and the recognition that the manuscript could be published in its current state.

Major comments:

Introduction:

1. *Line 51: The authors must add that Tubulin Tracker labeling binds only to polymerized microtubules (information that comes only in the discussion line 389) compared to microtubule visualization using immunostaining with anti-tubulin antibodies.*

We have now included a statement at the first mention of Tubulin Tracker (Line 192) that states that it only binds to polymerised tubulin. In lines 233 to 235, we have clarified that anti- β -tubulin can recognise monomers or oligomers of tubulin, as well as polymerised tubulin.

2. *Line 61: With the development of advanced microscopy techniques, we have gained tremendous knowledge about the subcellular architecture of the MTOC in the asexual stage of P. falciparum recently published in Simon et al. study (Preference number 17 in the current manuscript). However, it remains unknown if the MTOC is "present throughout the cell cycle" and in which form (presence or not of centrin1 protein and DNA-free intranuclear compartment); therefore, I would recommend the authors to soften their language and say it is "likely present...."*

The wording has been changed (Line 61), as suggested, to read "likely present throughout the cell cycle".

3. *Line 66-67: "In the parasite's equivalent of anaphase, the sister kinetochores are separated and pulled by microtubules towards the MTOCs (Reference: 18). The study cited by the authors confirmed the presence of 14 chromosomes in P. falciparum using 3-D reconstruction of EM sections and does not provide any data to test what mechanism drives chromosome separation during Plasmodium blood-stage mitosis. There are several models: (1) "kinetochore microtubules" pull chromosomes, (2) inter-bundle microtubule sliding moves chromosomes, (3) chromosomes slide through lateral interactions with microtubules, or (4) microtubules directly push chromosomes apart. All models have been discussed in a recent publication about mitosis in the Naegleria parasite (PMID: 35139359). Because future investigations are needed to address how chromosome segregation occurs during Plasmodium cell division, I would strongly suggest the authors rephrase lines 66-67 to reflect our current knowledge.*

We accept the reviewers' point. To avoid confusion, the wording has been changed to "...the sister kinetochores are separated and moved towards the MTOCs." (Line 67)

4. *Line 68-70: As the authors were carefully using the accurate term for the MTOC based on previously published data, I would encourage the authors to reword the following sentence by switching the "assembled" for "detectable" so the sentence will read "... and the nuclear microtubules remain detectable until the cell undergoes cytokinesis in the final stage of Schizogony" to not pass on overstatement from previous work published by another research team.*

We take the reviewer's point. The word "assembled" has been changed to "detectable", as suggested. (Line 69)

5. *Line 74: The authors did not reference their statement, which must be updated with the latest literature. In a recent study by Liffner and Absalon (PMID: 34835432), the authors observed subpellicular microtubules with typically 2–4 individual microtubules in each merozoite from segmented schizonts. More critically to the statement line 74, the authors of this study observed that subpellicular microtubules extend from the apical polar rings, along the length of the merozoite, and end at the basal complex using ultrastructure expansion microscopy combined with NHS ester and tubulin staining.*

We thank the reviewer for that suggestion. We have now updated the text and included reference to the Liffner and Absalon study (Line 73). We have also included a reference to the earlier EM study that supports the same conclusion.

6. Line 82-83: Reference is missing to support the statement that the gametocytes do not contain an apical ring, unlike merozoite.

We now include a reference ² to our previous serial block-face scanning electron microscopy study which supports the statement that gametocytes do not contain an apical ring (Line 82).

7. Line 93-96: The authors claim that "...nuclear microtubules are organized into a hemispindle-like a structure that drives elongation of the nucleus, reaching a maximum length of about 7 μ m in stage II/III of development," which is not entirely supported by their data. The authors demonstrated in this study that the same MTOC nucleates both nuclear and subpellicular microtubules, and they are both ablated by trifluralin treatment (line 104). Therefore they cannot exclude that a synergetic event could drive the reshaping of the nucleus during gametogenesis due to the extension of nuclear and subpellicular microtubules connected by a common MTOC.

We take the reviewer's point. We have now changed the text to read "...hemispindle-like structure that is associated with elongation of the nucleus." (Line 93)

Results:

8. Line 148-150: Although I agree with the authors' conclusion and their corresponding references, the authors might want to consider performing an additional experiment by performing ultrastructure expansion microscopy combined with NHS-ester and centrin staining as done in Simon et al. (reference 17) to determine whether the MTOC in asexual and sexual blood-stage parasites harbor an hourglass shaped-structure. This assay will also allow the authors to assess the architecture of the intranuclear compartment of the MTOC (devoid or not of chromatin), to validate the nucleation site of both nuclear and subpellicular microtubules (as the authors beautifully shown by TEM), and to define the organization of the nuclear microtubule (number of branches and length in the hemispindle-like?). Finally, the higher resolution obtained by expansion microscopy will allow redoing their measurement for Cen4- to CENH3 distance if the immunostaining works on expanded parasites.

We recognise that expansion microscopy experiments could add value to the current study; however, we believe that the combination of light microscopy and TEM employed in this work provides a good alternative to expansion microscopy. We have employed thin section transmission electron microscopy (TEM), serial section TEM and electron tomography to obtain views of the gametocytes at much higher resolution than can be achieved by expansion microscopy.

We believe that the suggested expansion microscopy study should be pursued as part of future investigations, rather than being included in the current manuscript, for the following reasons.

- i) Expansion microscopy is a technically very demanding technique, that requires access to some difficult-to-source bespoke reagents; and may take some months to establish in a new laboratory.
- ii) In this work, we have made a quantitative analysis of different phenomena, analysing more than 200 infected RBCs for some observations. Expansion microscopy is an inherently low throughput technique and it would be very difficult to achieve similar numbers of observations.
- iii) As the reviewer points out, in this work we have made a quantitative analysis of the length of, or distances between, different features. Such studies would not be possible with expansion microscopy, which produces samples that are not always expanded uniformly or in a highly controlled way.

9. Line 296-302: Based on figure 4A, NDC80 and CENH3 signals overlap in stage I and IV and not only in early-stage gametocytes, as mentioned in lines 296-297. It is unclear for what stage of development all Pearson's coefficients were calculated from. For instance, the "r" between NDC80 and Hoechst in stage IV should be close to 1 when it is reported to be 0.251. I recommend the authors show the graph of all Pearson's coefficients for each stage as an additional column on the right of panel A.

We thank the Reviewer for this suggestion. We have now undertaken the comparison of PfNDC80/PfCENH3, Hoechst/PfCENH3 and Hoechst/PfNDC80 fluorescence signals (and vice versa) (n = 20 cells) at each stage of

gametocyte development. In undertaking this analysis, we reached the conclusion that it would be more appropriate to present Mander's coefficients (M), generated using the JACoP FIJI plugin, which allows us to choose a region of interest rather than looking at co-occurrence across the entire image

The Mander's coefficients are presented in a new Table (Supp Table 1) along with a statistical analysis of the data (Supp Table 2) for different stages, to illustrate the trends in co-occurrence at different stages of gametocyte development. We have described the analysis protocol in the Methods section.

The analysis reveals some features that are not as obvious to the naked eye. For example, while the NDC80 does overlap with the Hoechst signal throughout gametocyte development (we added the NDC80/Hoechst combined channel in Figure 4A); the reverse analysis - Hoechst with NDC80 - increases to a moderate level in Stage III and then decreases. This is consistent with the expected location of kinetochores in a sub-region of the chromatin. Additional text has been added (Lines 301-307) to describe these changes.

Discussion:

10. Line 399-400: The authors conclude that the MTOC structure in gametocytes is similar in appearance to those found in dividing asexual/sexual blood-stage parasites. It is true if they had "using electron microscopy." Simons et al. study, I have nicely demonstrated using NHS-ester on expanded parasites that the asexual MTOC is an hourglass shape composed of two compartments: outer and intranuclear. Moreover, the intranuclear compartment is highly dense in protein devoid of chromatin.

We have now added the term "Using electron tomography,.. " to clarify that our statement is based on our electron tomography (ET) data – which gives a higher resolution view of the MTOC than can be achieved with expansion microscopy. It should be noted that every technique has its own potential for artefacts. The sample are fixed for ET, as they are for expansion microscopy; but expansion microscopy could also distort features due to inhomogeneous expansion. Given these concerns, it is pleasing that the different techniques give similar results with respect to the dumbbell shape of the MTOC. And our finding that dense packets of chromatin are located away from the MTOC also agrees with the expansion microscopy study showing that the region close to the MTOC is devoid of chromatin.

11. Line 411-413: The suggestion that the MTOC in male gametocytes bifurcated to form the basal body is exciting but not supported by the data on Pfcetrin-4 immunostaining due to lack of resolution.

As mentioned in the text, we observed that Pfcetrin-4-mCherry is associated with double puncta in a sub-set of the gametocytes (one in six). As detailed below (in response to Point 14), we have now reorganised Supp Figure 6A and 6B so that the subset of cells with double puncta is featured. We are able to distinguish the two puncta as separate objects because the average distance between the centres of the fluorescence peaks is greater than the resolution of the DV Elite microscope (i.e. 250 nm) used to collect these images. We have now included this quantitative analysis (Supp Figure 6D, n = 15 cells for each stage) and as insets (Supp Figure 6A and 6B) to highlight the two adjacent punctate features.

Minor comments:

Results:

12. Line 240-242: Supplementary Figure 5 C and F, beautiful live-imaging microscopy data demonstrated the localization of EBA1 foci at both ends of different types of intranuclear microtubules (mitotic and extended spindles). To avoid any confusion, I suggest the authors replace the word mitotic with intranuclear "in the nuclei of dividing schizonts located at either end of the mitotic spindles."

We thank the review for pointing this out. We have now changed "mitotic" to "intranuclear". (Line 246)

13. Line 245-247: In the absence of nuclear envelope marker, the authors should say chromatin instead for nucleus in the following sentence: " In contrast, the Pfcetrin-4-mCherry signal locates adjacent to the nucleus but does not overlap with the TubulinTracker signals (Supp Figure 5F)." Simon et al. (Ref 17) demonstrated that Pfcetrin-1 localizes exclusively in the outer nuclear compartment of the MTOC, and additional work is required to address Pfcetrin-4's exact localization in the MTOC of asexual blood-stage parasites.

We thank the review for pointing this out. We have now changed “nucleus” to “chromatin”. (Line 250)

14. Line 259-260: *The authors' resolution of light fluorescence microscopy is at best 200 nm; therefore, it is technically impossible to be sure that every single punctum of PfCentrin-4-mCherry is single double puncta. Interestingly, in a sub-set of the gametocytes (one in six), Pfcen-4-mCherry labeling is evident as two puncta (Supp Figure 6A, pink arrowheads, B)". I suggest the authors keep the "red" labeled pictures and remove panel B and the text referring to the figure in lines 259-260.*

We are a little confused by the reviewer's comments. We are not claiming that the single puncta are “single double puncta”. What we noticed is that in a sub-set of cells (possibly males), two adjacent puncta are evident. We discuss the possibility that these might be male gametocytes; though clearly more work is needed to clarify the nature of the two adjacent puncta. We have now analysed intensity profiles across the double puncta. The distance between the peaks increases from an average of ~400 nm in stage I/II to ~600 nm in Stage V (Suppl Fig 6D). In each case the separation is above the level of resolution of the DeltaVision deconvolution microscope used for this work.

Nonetheless, we accept that the previous Supp Figure 6A may have been confusing. We have therefore reorganised the figure as requested. We separated single and double puncta cells in two separate figures (Supp Figure 6A and 6B). We feel it is necessary to keep the panel that quantitates the frequency of single and double puncta.

We have also included a new Figure panel (Supp Fig 6D) which provides quantification of the distances between the puncta. We have added text to the figure legend. “The experiments were performed three times using different gametocyte batches. (C) The distance between the two centrin-4-mCherry puncta was quantified at different stages of development (n = 15 cells for each stage). The differences between the distances were analysed using an unpaired one-way ANOVA Turkey's test. I vs. II: NS = 0.9300; II vs. III: NS = 0.9976; III vs. IV: NS = 0.2429; IV vs. V: *p = 0.0109.”

In an effort to avoid confusion, we have also now altered the text to make it clear that more work is needed to test the suggestion that these double puncta represent bifurcation of the MTOC to form the basal body (Line 424-425).

15. Line 280: *Reference is missing, and it is unclear in which organism the authors are referring to vertebrates, unicellular eukaryotic cells, or parasites?*

We accept that this statement was confusing. The text has now been changed to:

“The attachment of chromatin to the nuclear microtubule bundle is reminiscent of the process that occurs during mitosis in *Plasmodium*, whereby the kinetochores assemble at specialized chromatin structures on each chromosome, called centromeres, and attach the sister chromatids to microtubule spindles to coordinate their migration into the daughter nuclei.” (Line 282-285)

16. Line 334. *The reference to figure 5 D is missing in the text*

Figure 5D illustrates the decreased aspect ratio upon trifluralin treatment. It is mentioned in the text - Line 328-329 “ ..resulting in a loss of the characteristic elongated shape (Figure 5A, D)..”.

Discussion:

17. Line 368: *The authors suggest interactions with IMC-associated alveolins stabilize the sub-pellicular microtubules. The alternative hypothesis supported by their data is that the polyglutamylation detected with the anti-polyE could stabilize the microtubules too.*

That part of the discussion refers to stage IIIb, at which point polyglutamylation is not yet evident by immunofluorescence. It is possible that there is already a low level of modification; but we prefer to leave the discussion of polyglutamylation till later in the Discussion when we discuss the stage IV gametocytes.

18. Line 394: *The authors wrote, "Trifluralin-treated gametocytes fail to assemble either sub-pellicular or nuclear microtubules," but they meant "and" (line 222)*

The text has been changed to “...fail to assemble both sub-pellicular and nuclear microtubules”. (Line 400)

19. Line 457: *References are missing. Simons et al. study demonstrated that hemispindle microtubules do not interact with centromeric proteins using CenH3 as a marker. They also used live-imaging to determine the timing of intranuclear microtubules organization for the first time.*

We apologise. The wrong reference was used here. It has now been corrected to Simon et al³. (Line

465) General comments for Figures:

20. *Although the authors thoughtfully displayed separate channels in addition to the merged images. I would recommend the authors to rethink the choice of their representative colors for immunostaining and live-imaging regardless of the actual color of the fluorescent protein to make their figures accessible for color-blindness and to increase interpretation for everyone (the cyan color for TT staining made it challenging to interpret merges with green and avoid red, especially in combination with green).*

To help the authors see the below-suggested combination of colors: • Magenta/green/blue; • Magenta/Yellow/Cyan • Red/Cyan/Yellow

We appreciate the concerns regarding the use of green and red as a colour combination and we did explore other possibilities in the preparation of this manuscript. However, we ran into difficulties in the panels in which 4 colours are used, *i.e.*, 2 fluorescent protein channels, plus tubulin tracker, plus the nuclear stain. It is very difficult to find a combination of non-red/green colours that works for 4 colour panels. The overlay of such colours appears white; and we needed the greyscale channel for the Hoechst. For most of the panels, we have therefore chosen to provide individual channels as an alternative means of enabling colour blind individuals to see the different features. We have also undertaken quantitative analyses that help to interpret the data. For some panels where there were only 2 colour channels used or where there were limited overlapping features, eg Fig 2D, Supp Fig 4,5,7,8, we have used non-red/green colour schemes.

If the Editor insists, we can change the colour schemes; but we feel that this will detract from the ease of interpretation of the data.

21. *Figure 4. Panel A: Stage V is missing in the panel; could the authors elaborate on why?*

Fig 4A shows the contraction of *PfNDC80* and *PfCENH3* back to the MTOC. This has already occurred by Stage IV, with no further reorganisation in stage V, as is evident in Fig 4B-D. Additional representative images *PfNDC80* and *PfCENH3* were provided in Supp Fig 9 and 10, including stage V. We therefore did not include a Stage V representative cell in Fig 4 so that the figure would be less crowded.

Nonetheless, we appreciate the reviewer's point, and have now included a stage V example in Fig 4A.

22. *Figure 4. Panel B/C/D: The authors might consider drawing a horizontal line around 0.250 μm , marking the resolution indicating that measurements below the line two foci distant of 250 nm or lower will be seen as one focus. See 23 below.*

23. *Figure 5: For Panel B/C: Same comments as for Figure 4*

At the reviewer's request, we have now added the horizontal line at 0.25 μm to the graphs to illustrate the limit of resolution.

24. *Figure 5: In addition, the authors might consider adding a schematic on the top of each graph to explain what is "contour length" and "Aspect ratio"? It will increase the likelihood for another lab to reproduce the study's data if there are no ambiguities on how the cellular measurements were done.*

We have now added a series of schematics in Supp Fig 13 to explain how contour length, maximum length, maximum distance and aspect ratio were measured.

25. *Figure 6: I compliment the authors for creating a data/model figure. I have a few suggestions. To avoid the combination of Red/green, the parasite could be gray. The column with the RBC present could be smaller to gain space for an additional zoom on the parasite drawing to see better the details in the drawing. The caption could take the entire width of the figure and place it in the bottom to free space for zoom on the parasite like done for other stages.*

As suggested, the model in figure 6 has been redrawn so that the parasite is grey and the zoom panels enhanced and presented for all stages.

27. *Supp Figure 03: Panel B: The loading control for the 3D7 sexual stage is pretty low, and therefore it is difficult to conclude anything regarding EB1 expression level.*

We thank the reviewer for this comment. We have now replaced the figure panel with a new data set showing asexual and gametocyte stages of the EB1-GFP transfectants probed with the anti-GFP and anti-PfERC antibodies. EB1-GFP is detected at the expected molecular weight in the transfectants but not the in the parent 3D7 line, i.e. the negative control.

28. *Supp Figure 04: Panel D: as for panel B, it should read anti-GFP and not EB1-GFP. In my understanding, all figures from panels B and D are from immunostaining, therefore "or endogenous PfEB1-GFP labeling" in lines 68-69 should be removed*

We used two different microscopes to generate the immunofluorescence image data. A Zeiss Elyra LSM880 confocal microscope was used to image anti-GFP binding to EB1-GFP (Supp Figure 4B). A DeltaVision Elite deconvolution microscope was used to acquire the images of the weaker remnant EB1-GFP fluorescence (Supp. Figure 4D). We have now updated the methods section to make it clear which microscope was used to collect which datasets. We have added clarifying text to the figure legend: "The PfEB1-GFP signal was amplified using anti-GFP (B) or measured directly as the remnant PfEB1-GFP signal (D)."

29. *Supp Figure 05: Panel B: Could the authors flip the blot picture vertically, starting with 3D7 samples first as for all other blots in other figures and Panel E?*

The Western blot images has been flipped to show the 3D7 samples on the left.

30. *Supp Figure 6: A title reflecting the data rather than the experimental design would be best.*

The title has been changed to: "Reorganisation of nuclear microtubule marker, PfEB1-GFP, relative to gametocyte centriolar plaque/MTOC marker, Pfcentrin-4, and evidence for two Pfcentrin-4 puncta in a sub-set of gametocytes."

31. *Supp Figure 11: Panel G: The loading control is missing for the anti-PolyE blot.*

The blot in Supp Fig 11G is reprobed to see the bands for the 3 different antibodies. The full-length blot for anti-PfERC is now provided adjacent to the anti-tubulin and anti-polyE blots, to make this clearer.

The figure legend has been modified to make this clear "Control and trifluralin-treated gametocytes were harvested on Day 7 of treatment and subjected to Western transfer. The blot was probed with anti-polyE, anti-13 tubulin (anti-13 tub) and anti-PfERC (loading control), with stripping between antibody applications."

32. *Supp Figure 12: The title reads, "Trifluralin treatment...disrupts cellular remodeling, as..." However, it is vague and suggests the author write something like "chromatin spatial organization" instead.*

The title has been changed to read: "Trifluralin treatment of ring stage gametocytes disrupts cellular remodeling, as revealed by kinetochore and centromere markers."

Reviewer #2 (Remarks to the Author):

The manuscript present by Li et al from Leann Tilley's group is a very illuminating and interesting study illustrating how some of the components of mitotic machinery are used by the human malaria parasite P falciparum in the cellular shape and chromatin reorganisation. P falciparum undergoes various cellular organisation changes during its gametocyte development. The authors show how the cytoplasmic and nuclear microtubule orient in these different stages and how the microtubule organising centre is organised during these different stages even if they are not undergoing mitosis. This is the originality of this work. The authors have used some cutting edge cell biology techniques like 3D electron microscopy and real time live cell imaging which are quite challenging to do in the human parasites. The different markers used in the study are centrosomal marker like centrin 4 and 1, kinetochore marker Ndc80 and microtubule end binding protein EB1. Some of these markers have been studied during mitotic stages by our group and by others (Roques et al 2018; Zeeshan et al 2020; Simon et al 2021, Raspa and Brochet 2022) and hence that is not surprising what they observe but since the study is focussed on the non-mitotic cell and hence it is original in that aspect. The study is bit diluted by the recent study by Raspa and Brochet, PLoS Pathogens 2022, where they have shown that microtubule localisation at the sub-pellicle region of the human parasite using expansion microscopy. However saying that, the live cell imaging of the Ndc80 and EB1 is exemplary in this manuscript and gives a better picture of the mitotic machinery in this study. This will be very useful for the community interested in the mitosis and sexual development both in human parasite but also to scientist working on the other pathogens belonging to Apicomplexa and other showing atypical microtubule dynamics.

The authors also substantiate their study by using trifluralin by depolymerising the microtubules and demonstrate that by this treatment blocks the chromatin and kinetochore organisation and thereby IMC positioning in gametocytes.

The study is solid though some of these molecules have been studied in depth before as mitotic markers. It would have been interesting if some of functional data on some these molecules were attempted to show if they affect the gametocyte organisation and at what stages do they affect it. As it has been shown that centrin 4 is not required for asexual stage multiplication and hence it will interesting to know if the centrin 4 affect the non-mitotic assembly during gametocyte. The same will be informative if it was demonstrated what happens with EB1 as in Toxo it is not required for asexual multiplication.

I really enjoyed reviewing the manuscript and it is an excellent study and some outstanding multimodal electron microscopy and real time live cell imaging is present for gametocyte. There is lot of data in the supplementary about the asexual stages which is very informative for the community. Overall this study will be of great interest to the scientist working on MTOC and mitosis in Plasmodium and other non-model eukaryotes.

Rita Tewari

We thank Dr Tewari for her very kind comments on our manuscript. We appreciate that a number of expansion microscopy studies appeared while this work was in preparation for publication. However, this is the first study that reports the presence of nuclear microtubules in non-mitotic gametocytes. We are pleased that Dr Tewari appreciates the general interest for scientists working on MTOC and mitosis in *Plasmodium* and other non-model eukaryotes.

We appreciate the reviewer's request for functional data on some of the molecules involved in the nuclear microtubule structure. We agree that such studies are warranted but suggest that these studies require a very large body of work that is best addressed in future manuscripts.

Reviewer #3 (Remarks to the Author):

In this manuscript Li and colleagues describe an enigmatic structure resembling a mitotic spindle during Plasmodium micro- and macro-gametocytogenesis, two developmental stages that are not known to undergo mitotic division. However, three rounds of endomitosis quickly happen following microgametocyte activation by a decrease of temperature of at least 5C. In the absence of genome replication and mitosis, the authors propose that this process is used by gametocytes to drive their characteristic cellular elongation and to reorganise chromatin. To describe this structure, the authors combine various imaging approaches including 3D electron microscopy and fluorescence microscopy with known markers of the mitotic spindle, the kinetochore or the centromere.

1. The idea that Plasmodium falciparum gametocytes use the mitotic machinery for cell elongation and chromatin organisation is tantalizing and would represent an important discovery. Gametocytogenesis attracts a growing

interest but has also be well scrutinised by electron microscopy and such structure has never been observed nor described, which is relatively surprising. Therefore, I think that the authors should make 100% sure they are not observing an artefact of premature activation of microgametocytes due to unwanted temperature drops, which are in our hands frequently observed particularly at pH >7.2. In the comments below we note some observations that raise some concerns and propose experiments to address them.

We thank the reviewer for recognising the importance of this study. We would also like to point out that microgametocytes can only be activated once they are mature, i.e. stage V. The gametocytes used in this study are highly synchronised, as quantitated in Supp Fig 11D. We observed nuclear microtubules only in Stage II/III gametocytes. By stage V the microtubule bundles appeared to have been disassembled. Therefore, we are confident we are not looking at premature activation.

The reviewer suggests that nuclear microtubules have never been described by EM. As pointed out in the manuscript, early ultrastructural studies of *P. falciparum* gametocytes provided single section TEM images of intranuclear bodies with a hemispindle appearance, emanating from a region of electron-dense plaque in an enlarged nuclear pore^{4, 5}. (Line 453)

The failure of previous fluorescence microscopy studies (including our own) to recognise the presence of nuclear microtubules likely arises from the fact that these studies did not include a nuclear marker. In the absence of such a marker it is very difficult to distinguish nuclear microtubule bundles from subpellicular microtubule bundles.

Major comments

2. In figure 1 it is not clear how many cells were analysed.

Figure 1 presents the results from different electron microscopy techniques. Panels A-C and Suppl Fig 1 represent thin-section TEM data. These data represent a selection from hundreds of sections that were examined in the course of this study. For example, line 178 states “The nuclear microtubules have the same average diameter as the sub-pellicular microtubules (26 + 3 nm (n = 84) and 27 + 4 nm (n = 109), respectively) and are separated by an average distance of 15 + 4 nm (n = 79) resulting in an average bundle diameter of 320 + 25 nm (n = 50).”

The EM serial section data (Panels D-F) and electron tomography (Panels G-K) are very technically demanding techniques. The data presented are representative to 2 or 3 examples collected for each stage examined. We also employed Array Tomography surveys of stage I-V gametocytes; a technique that is amenable to higher throughput. Suppl Fig 2 presents the results from 10 cells for each developmental stage.

3. Could the authors use known markers of male and female gametocytes to ascertain that their observations hold true for both sexes?

To our knowledge, male/female-specific markers are only available for late stage (IV/V) gametocytes. The nuclear microtubules are only present in early (stage II/ III) gametocytes.

Due to this technical limitation, we took a different approach. We used NLS-mCherry and Hoechst signals to mark the nucleus and looked for the Tubulin Tracker-labelled microtubules in this compartment. As stated (Line 201), we observed nuclear microtubules in 100% of the 249 stage II/III gametocytes examined, indicating that the nuclear microtubules are present in both male and female gametocytes.

In an effort to make this clearer to the reader we have now added text to this section to explain the strategy, given the lack of antibody reagents that can be used as sex-specific markers for early stage gametocytes (Line 198).

4. In figure 2, could the authors confirm with an ER marker such BiP the proposed correlation between the endoplasmic reticulum, the nascent IMC and the microtubule organisation? It would be important to quantify particularly if the number of cells analysed in figure 1 is limited.

We appreciate the reviewer’s concern; but would like to point out that juxtaposition of the ER and IMC has been reported previously. As indicated in the text, in the early stages of merozoite formation in *P. falciparum*⁶ and in

endomitotic *T. gondii*⁷, ER exit sites are located close to the centriolar plaque structure; and are assumed to generate the membrane material for the apical organelles and the IMC.

We have also previously reported a continuum between the ER and the IMC using super-resolution imaging of the ER marker, anti-*Pf*ERC, and antibodies recognising the IMC marker, GAP50-GFP⁸. We have now included a reference to this study in the manuscript. (Line 380)

We do not feel that it is valuable to repeat those experiments for this manuscript, but if the Editor/ reviewer insists, we are prepared to consider such experiments.

6. It would be essential to confirm that the observations made in figure 2 hold true for both male and female gametocytes using specific markers. The authors show parasites expressing NLS-mcherry. However, it seems that that expression of this reporter protein is under the control of the hsp86 promoter. According to transcriptomics studies available on PlasmoDB, hsp86 promoter is up to 10 times more active in microgametocytes. It is thus possible that the authors have been focusing their analysis on microgametocytes. Similarly, in figure 3 and 4 it would be essential to ensure that the measurements hold true for both macro- and microgametocytes.

As described above, to our knowledge, male/female-specific antibody reagents are only available for late stage (IV/V) gametocytes; while nuclear microtubules are only present in early (Stage II/ III) gametocytes. Moreover, we have examined hundreds of well synchronised stage II/III gametocytes and observed nuclear microtubules in 100% of the cells. Male gametocytes represent only ~20% of the gametocytes in a culture and therefore we are confident that the nuclear microtubules are present in both male and female gametocytes.

7. Finally, if only microgametocytes show this structure, the authors should ensure that this observation is not linked to unwanted premature and suboptimal activation of microgametocytes due to experimentally inherent fluctuations in temperature. It was previously shown that in Plasmodium falciparum, CDPK4 is essential for the formation of the mitotic spindle during microgametogenesis but that mutant gametocytes lacking cdpk4 are nevertheless elongated (Kumar et al, 2021, mbio). I would strongly recommend to the authors to document the formation of these structures resembling a mitotic spindle during gametocytogenesis of mutants lacking cdpk4 or in presence of an inhibitor of CDPK4 such as compound BK11294. This should rule out that the authors are describing an experimental artefact.

As described above the nuclear microtubules are observed in 100% of stage II/III gametocytes. We thank the reviewer for their suggestion regarding CDPK4 knockout parasites. This is an interesting suggestion. However, we feel that that should be the subject of a further publication. The Kumar et al publication shows that *Pf*CDPK4 is dispensable for asexual replication. This means that the asexual mitotic spindle is not affected. We are proposing that the nuclear microtubules present in early stage gametocytes use the same machinery as for the asexual blood stage mitotic spindle. Therefore, we would not anticipate that CDPK4 knockout parasites would exhibit a defect in the gametocyte nuclear microtubules.

Minor comments

8. In figure 4, it is not clear how the authors define the relative movement of Ndc80, CenH3 to the centriolar plaque(s) in the absence of a centrin staining and with a bimodal distribution of CenH3 and Ndc80.

As suggested by reviewer 1, we have now included schematic diagrams (in Supp Fig 13) to illustrate what is being measured and how. We apologise for the confusion. The quantification of the CenH3 to centriolar plaque distances are based on an analysis of data for the *Pf*centrin-4-mCherry/GFP-*Pf*CENH3 co-transfectants (Figure 4E), rather than the *Pf*NDC80-mCherry/GFP-*Pf*CENH3 co-transfectants (Figure 4A). We now highlight which cell line is used in the legend to Figure 4D.

9. The authors propose that this microtubule spindle is important for the elongation of the cells. The images shown here suggest that the orientation of the spindle is indeed sagittal to the longest length of gametocytes but this is not quantified. Could it also be a consequence of elongation rather than a cause?

We do suggest in the manuscript that the mitotic machinery is repurposed to drive cellular elongation. However, this suggestion is based on our finding that the MTOC also nucleates the sub-pellicular microtubules. We propose that this, in turn, elongates the cell.

In an effort to make this clearer, we have reworded the final statement to say “ The MTOC and associated microtubule network are non-mitotic, instead initiating and co-ordinating IMC formation, driving nuclear and cellular elongation and potentially promoting cellular rigidification”. (Line 470-472)

10. *A link between mitotic MTOC and cell polarity has already been described in Toxoplasma (see for example the review from Harding and Frischknecht (Trends in parasitology, 2020) and would be worth mentioning.*

We thank the reviewer for this suggestion and have now included this reference (Line 381-382).

References

1. Bannister LH, Hopkins JM, Fowler RE, Krishna S, Mitchell GH. A brief illustrated guide to the ultrastructure of *Plasmodium falciparum* asexual blood stages. *Parasitol Today* **16**, 427-433 (2000).
2. Parkyn Schneider M, *et al.* Disrupting assembly of the inner membrane complex blocks *Plasmodium falciparum* sexual stage development. *PLoS Pathog* **13**, e1006659 (2017).
3. Simon CS, *et al.* An extended DNA-free intranuclear compartment organizes centrosome microtubules in malaria parasites. *Life science alliance* **4**, (2021).
4. Sinden R. Gametocytogenesis of *Plasmodium falciparum* in vitro: an electron microscopic study. *Parasitology* **84**, 1-11 (1982).
5. Sinden RE, Canning EU, Bray RS, Smalley ME. Gametocyte and gamete development in *Plasmodium falciparum*. *Proc R Soc Lond B Biol Sci* **201**, 375-399 (1978).
6. Bannister LH, Hopkins JM, Fowler RE, Krishna S, Mitchell GH. Ultrastructure of rhoptry development in *Plasmodium falciparum* erythrocytic schizonts. *Parasitology* **121 (Pt 3)**, 273-287 (2000).
7. Francia ME, Striepen B. Cell division in apicomplexan parasites. *Nat Rev Microbiol* **12**, 125-136 (2014).
8. Dearnley MK, *et al.* Origin, composition, organization and function of the inner membrane complex of *Plasmodium falciparum* gametocytes. *J Cell Sci* **125**, 2053-2063 (2012).

Reviewer comments, second round review

Reviewer #1 (Remarks to the Author):

The authors did an absolutely fantastic job answering my major and minor comments. I appreciate how the authors went beyond my request and I personally learn a lot from the authors and the revision process. The author's work is contributing to advancing our understanding of the atypical and non-canonical microtubule biology in Plasmodium parasites and in the eukaryotic cells at large.

Reviewer #3 (Remarks to the Author):

This new version of the manuscript is much improved and clearer. As the authors now more clearly indicate that hemispindles are observed in 100% of stage II-III gametocytes only, I assume these observations cannot be linked to premature activation of microgametocytes.